# Ultrafast complete dechlorination enabled by ferrous oxide/graphene oxide catalytic membranes via nanoconfinement advanced reduction

Qian Xiao[1,2,3], Wanbin Li[4] ✉, Shujie Xie[1], Li Wang[1] & Chuyang Y. Tang[1,5] ✉

Chlorinated organic pollutants widely exist in aquatic environments and threaten human health. Catalytic approaches are proposed for their elimination, but sluggish degradation, incomplete dechlorination, and catalyst recovery remain extremely challenging. Here we show efficient dechlorination using ferrous oxide/graphene oxide catalytic membranes with strong nanoconfinement effects. Catalytic membranes are constructed by graphene oxide nanosheets with integrated ultrafine and monodisperse sub-5 nm nanoparticles through simple in-situ growth and filtration assembly. Density function theory simulation reveals that nanoconfinement effects remarkably reduce energy barriers of rate-limiting steps for iron (III)-sulfite complex dissociation to sulfite radicals and dichloroacetic acid degradation to monochloroacetic acid. Combining with nanoconfinement effects of enhancing reactants accessibility to catalysts and increasing catalyst-to-reactant ratios, the membrane achieves ultrafast and complete dechlorination of 180 $\mu g\,L^{-1}$ dichloroacetic acid to chloride, with nearly 100% reduction efficiency within a record-breaking 3.9 ms, accompanied by six to seven orders of magnitude greater first-order rate constant of 51,000 $min^{-1}$ than current catalysis. Meanwhile, the membranes exhibit quadrupled permeance of 48.6 $L\,m^{-2}\,h^{-1}\,bar^{-1}$ as GO ones, because nanoparticles adjust membrane structure, chemical composition, and interlayer space. Moreover, the membranes show excellent stability over 20 cycles and universality for chlorinated organic pollutants at environmental concentrations.

Haloacetic acids (HAAs) is an abundant group of disinfection by-products (DBPs) in water, whose maximum level could reach up to 600 $\mu g\,L^{-1}$ in chlorinated drinking water[1]. Dichloroacetic acid (DCAA), a dominant HAA, is one of the most widely produced DBPs in chlorinated water[2]. Inhalation or ingestion of DCAA can harm liver organs, as well as neurological and reproductive systems[1,3]. It has been classified as a probable carcinogen to humans by the International Agency Research on Cancer[3,4]. China and the United States have

[1]Department of Civil Engineering, The University of Hong Kong, Hong Kong SAR 999077, China. [2]State Key Laboratory of Pollution Control and Resource Reuse, College of Environmental Science and Engineering, Tongji University, Shanghai 200092, China. [3]Shanghai Institute of Pollution Control and Ecological Security, Shanghai 200092, China. [4]Guangdong Key Laboratory of Environmental Pollution and Health, College of Environment and Climate, Jinan University, Guangzhou 511443, China. [5]Materials Innovation Institute for Life Sciences and Energy (MILES), HKU-SIRI, Shenzhen 518000, China. ✉e-mail: gandeylin@126.com; tangc@hku.hk

regulated a maximum contamination level of $50 \mu g L^{-1}$ for DCAA and $60 \mu g L^{-1}$ for the sum of five HAAs including DCAA, monochloroacetic acid (MCAA), monobromoacetic acid (MBAA), dibromoacetic acid (DBAA), and trichloroacetic acid (TCAA) in drinking water[2,5]. Furthermore, DCAA is of good water solubility, which facilitates its mobility in aquatic environments, rendering it difficult to be eliminated.

Several methods, including electrocatalytic[6,7], photochemical[8–10], and bulk catalytic[11,12] technologies, have been proposed for decomposition of HAAs. However, these technologies often suffer from sluggish degradation process. Additionally, many existing methods result in incomplete dehalogenation (e.g., promoted formation of MCAA[13]). Sulfite (S(IV))-based advanced reduction processes (ARPs) are highly efficient in removing refractory halogenated organic contaminants and inorganic oxyanions, from bench to pilot studies[14–18]. The ARPs generate reactive reducing species, such as sulfite radicals, and facilitate the cleavage of carbon-halogen bonds (i.e., C-Cl)[19,20]. For catalysis, however, the decomposition rate (~0.15 s$^{-1}$) of transition metal ions-sulfite complexes was relatively slower[16,21–24]. The energy-intensive separation (e.g., centrifugation)[25] for activation agents and the leached secondary pollutants (e.g., metal ions)[26] also hamper the practical application.

Catalytic membranes can integrate dual functions of filtration and catalysis[27–30]. Membranes can harness catalysts and spatially confine reactants and active sites at the micrometer- and nanometer-scale, which offer the potential to enhance catalytic performance; meanwhile, catalysis can facilitate decomposition of contaminants that cannot be effectively rejected by membranes. To date, no study has combined ARPs and membrane separation, despite that great progress of catalytic membranes have been made in gas-phase reaction, organic degradation, as well as bulk/fine chemicals[31,32]. Current catalytic membranes have been typically fabricated by anchoring catalysts onto ceramic and polymeric membranes through blending, surface coating, and bottom-up synthesis[33]. These approaches often led to the dropping and deactivation of catalysts, ineffective dispersion of catalyst particles, and degradation of catalytic membranes[33,34]. Graphene oxide (GO) surfaces contain an abundance of functional groups, thereby facilitating catalyst loading and mitigating nanoparticle (NP) aggregation. We, thus, posit designing ultrafine iron-based NP/graphene oxide composite membranes for advanced reduction of HAAs, which provide nanoconfinement effects to achieve reduced activation energy for the formation of reducing radicals for DCAA dechlorination. Furthermore, the reductants, including S(IV) and contaminants, must undergo nanoconfined transport through the catalyst-loaded interlayer channels of the iron-based NP/graphene oxide composite membranes. This greatly enhances the reductant-catalyst contact probability compared to the bulk solution based catalysts. Additionally, the catalysts and reactions will be nanoconfined in the interlayer transport nanochannels of GO membranes, which will substantially increase the catalyst-to-reactant ratios in confined reaction regions and then promote contaminant degradation.

In this study, we report efficient dechlorination of DCAA through constructing catalytic membranes by deposition of ultrafine FeO NPs on GO nanosheets. Through simple in-situ growth, monodispersed sub-5 nm FeO NPs can be deposited on oxygen functional groups of GO. The Fe/GO membranes possess ultrafast degradation performance of DCAA to chloride, with a nearly 100% removal within a record-breaking 3.9 ms, accompanied by a six to seven orders of magnitude greater first-order rate constant of 51,000 min$^{-1}$ than conventional processes (0.0013 – 33.3 min$^{-1}$), which can be attributed to enhanced reactants accessibility to catalytic sites, increased catalyst-to-reactant ratios in reaction regions, and reduced activation energy induced by nanoconfinement effects for FeSO$_3^+$ dissociation into sulfite radicals and for DCAA dechlorination. Meanwhile, the Fe/GO membranes give a quadruple water permeance of $48.6 L m^{-2} h^{-1} bar^{-1}$ as GO ones, due to the adjustment of FeO NPs to oxidation degree,

interlayer space, and membrane structure. Moreover, the prepared membranes exhibit excellent stability with maintained performance over 20 cycles and great universality for dehalogenation and reduction of other HAAs (>97%) and chlorinated organic pollutants (>90%).

## Results

### In-situ growth of Fe/GO nanosheets

We synthesized the Fe/GO composite nanosheets through in-situ growth of FeO NPs by introducing a 1.5 mmol L$^{-1}$ FeCl$_3$ solution (20 mL) to the GO suspension (20 mg L$^{-1}$, 150 mL) and reduction of Fe$^{3+}$ by NaBH$_4$ (1.6 M, 75 μL). The electronegative oxygen functional groups of GO adsorbed Fe$^{3+}$ ions, and then FeO NPs would be preferentially and heterogeneously nucleated at active sites under NaBH$_4$ reduction. FeO NPs were obtained by introducing NaBH$_4$ to reduce the Fe$^{3+}$ ions that are adsorbed onto the GO. Transmission electron microscopy (TEM) and scanning transmission electron microscopy (STEM) images show monodispersed sub-5 nm NPs with a high number density of $\sim 2 \times 10^5 m^{-2}$ on the Fe$_{c1.5}$/GO nanosheets (Fig. 1a–d and Supplementary Fig. 1). High-resolution TEM image indicated the presence of lattices with space of 2.54 and 2.15 Å, which are assigned to the (111) and (200) facets of FeO, respectively (Fig. 1c, inset). Electron diffraction implied that the NPs were FeO with exposed facets of (111), (200), (220), and (222) (Fig. 1d, inset). Energy-dispersive X-ray spectroscopy (EDS) analysis revealed the presence of elemental Fe and O (Fig. 1e, f), indicating successful in-situ growth and uniform dispersion of FeO NPs throughout the GO. As shown in the atomic force microscopy (AFM) image (Fig. 1g), the GO nanosheets were single-layered with a thickness of 1.0 nm, while the single-layered Fe$_{c1.5}$/GO nanosheets had slightly rougher surface than GO due to the attachment of ultrasmall NPs (Fig. 1h).

X-ray photoelectron spectroscopy (XPS) analysis was carried out to study the chemical binding states. Relative to GO with a C/O ratio of 2.42, the Fe$_{c1.5}$/GO nanosheets exhibited a higher C/O ratio of 3.27, despite the existence of O in FeO, suggesting the reduction of GO after in-situ growth of FeO. In addition to the two main peaks of C and O in the XPS spectrum, a new Fe peak was observed in the XPS spectrum of Fe$_{c1.5}$/GO, with the atomic content of 0.75%. High-resolution XPS spectra indicated that Fe existed as FeO and Fe$_2$O$_3$, with contents of 53.6% and 46.4%, respectively (Fig. 1i). No obvious peaks were observed for zero-valent iron. High-resolution C 1 s spectra of GO and Fe$_{c1.5}$/GO could be deconvoluted into three peaks of $sp^3$ hybridized carbon atoms at 284.8 eV, hydroxyl and epoxy at 287.1 eV, and carbonyl and carboxyl at 288.2 eV (Fig. 1j)[35–37]. Consistent with the variation in the C/O ratio, the contents of C-O and C=O declined from 36.1% and 12.5% for GO to 22.4% and 10.2% for Fe$_{c1.5}$/GO, respectively, suggesting the reduction of GO after in-situ growth. Notably, the GO after immersion in NaBH$_4$ (reducing agent) solution without FeCl$_3$ displayed only slight variations in the C/O ratio and oxygen-containing group content (Supplementary Fig. 2). This phenomenon implied that FeCl$_3$ might play an important role in promoting the reduction of GO, potentially by the involvement of Fe$^{3+}$ as electron shuttling, which was facilitated through the electrostatic adsorption between Fe$^{3+}$ and negatively charged GO sheets. Raman spectroscopy was used to investigate the regularity of nanosheet structures. Obviously, the spectra of GO and Fe$_{c1.5}$/GO showed two peaks of D (1322 cm$^{-1}$) and G (1592 cm$^{-1}$) bands, which were ascribed to the breathing vibration of defect/disordered $sp^3$ and the in-plane stretching vibration of original $sp^2$, respectively[38]. Compared to GO with ID/IG ratio of 1.32, the Fe$_{c1.5}$/GO nanosheets were slighter disorder with a lower ID/IG ratio of 1.21 owning to the reduction, which agreed with the XPS results.

Besides Fe$_{c1.5}$/GO prepared with a FeCl$_3$ concentration of 1.5 mmol L$^{-1}$, the Fe/GO nanosheets were synthesized with other precursor concentrations. For the nanosheets prepared with FeCl$_3$ concentrations of 0.5 and 1.0 mmol L$^{-1}$, i.e., Fe$_{c0.5}$/GO and Fe$_{c1.0}$/GO, the Fe atomic content was 0.40% and 0.60%, respectively, lower than that of

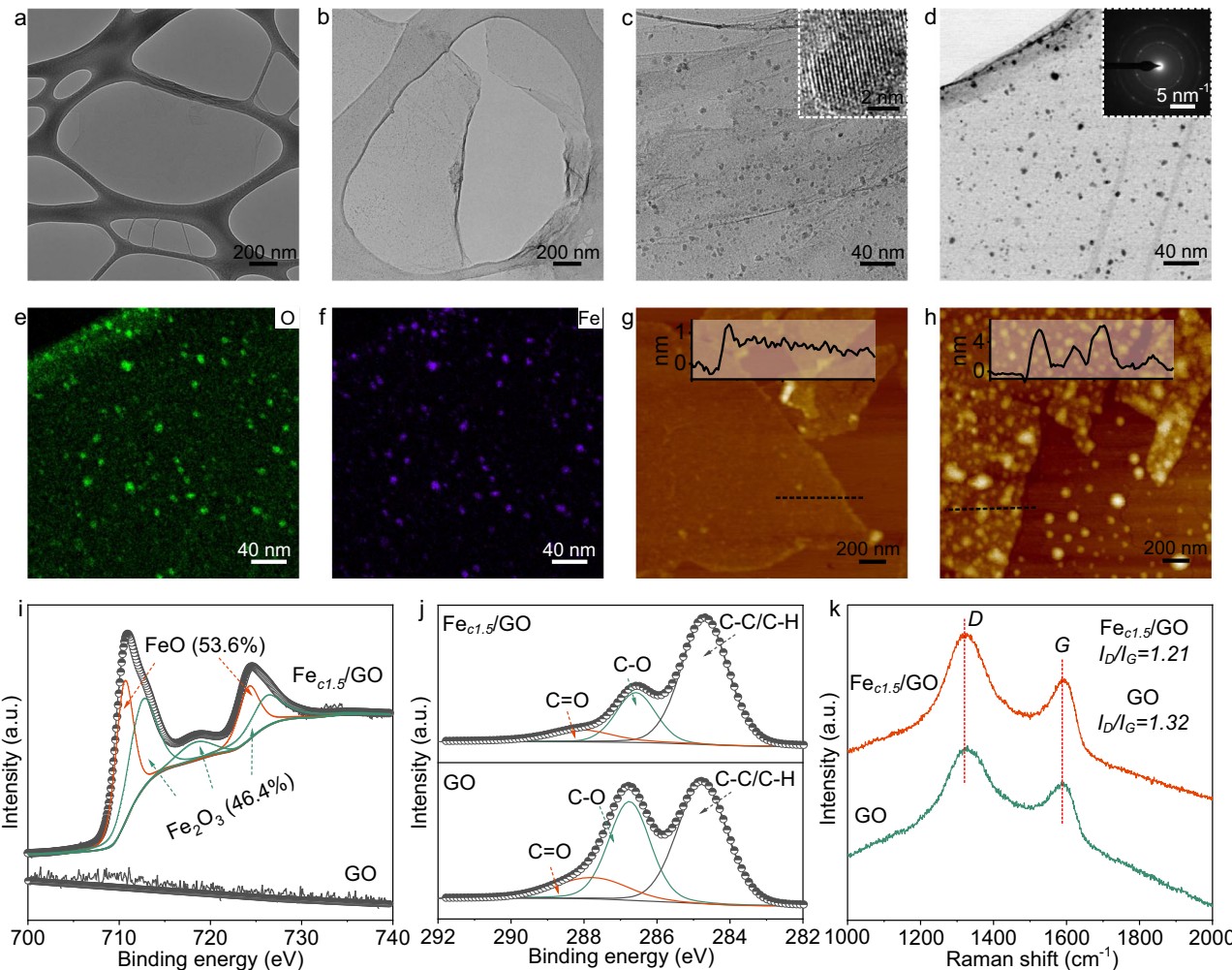

**Fig. 1 | Characterizations of Fe/GO nanosheets. a** TEM image of a typical GO nanosheet. **b, c** TEM image of a typical Fe$_{c1.5}$/GO nanosheet. Inset presents the TEM image of FeO NPs on nanosheet. **d** STEM image of a typical Fe$_{c1.5}$/GO nanosheet. Inset presents electron diffraction. **e, f** EDS mapping images of the Fe$_{c1.5}$/GO nanosheet. **g, h** AFM images of the GO and Fe$_{c1.5}$/GO nanosheets. Inset presents height profiles. **i,** Fe $2p$ XPS spectra of GO and Fe$_{c1.5}$/GO. The proportions of Fe(II) and Fe(III) were estimated based on the peak areas. **j** C $1s$ XPS spectra of GO and Fe$_{c1.5}$/GO. **k** Raman spectra of GO and Fe$_{c1.5}$/GO. a. u. represents arbitrary unit.

Fe$_{c1.5}$/GO. As expected, the C/O ratio, oxygen-containing group content, and structural regularity fell between those of GO and Fe$_{c1.5}$/GO, as identified by XPS and Raman results (Supplementary Figs. 3–5). For the nanosheets synthesized with concentrations > 1.5 mmol L$^{-1}$, the excessive loading induced the formation of defects in the Fe/GO membranes, as demonstrated below.

**Construction of Fe/GO membranes**

After in-situ growth of FeO NPs, the Fe/GO nanosheets were filtrated onto a polyvinylidene difluoride substrate with a pore size of 0.22 μm to fabricate the Fe/GO membranes (Supplementary Fig. 6). Photographs indicated that the nanosheets uniformly covered the substrates, and membranes with higher FeO loadings appeared darker (Supplementary Fig. 7). The GO and Fe$_{c1.5}$/GO membranes were uniform, defect-free, and ultrathin, with the thickness < 100 nm (Fig. 2a–d). Some wrinkles appeared on the GO membrane due to nanosheet flexibility, and the wrinkles of the Fe/GO membranes were weaken as loading increased (Fig. 2a, c, e, f and Supplementary Fig. 8). This phenomenon was explained by that the FeO growth reduced the flexibility of nanosheets, which might be beneficial to immobile the mass transfer channels during filtration under pressure and then maintain permeation properties. We prepared the Fe/GO membranes with higher loadings over 1.5 mmol L$^{-1}$ (Supplementary Fig. 9), but

many defects were generated in these membranes, because small nanosheet flexibility and excessive FeO loading caused irregular stacking. Top and cross-sectional view EDS mapping images revealed the homogeneous iron distribution throughout the Fe/GO membranes (Fig. 2g, h).

For GO membranes, the interlayer space is not only dominant transport channels for permeation, but provides sites for the confined reaction[39]. We studied the crystalline structure and interlayer space of the prepared membranes. From the X-ray diffraction (XRD) patterns (Fig. 2i), it could be seen that the diffraction peak for (002) plane of the GO membrane was located at 10.6°, which was consistent with previously reported value[40]. After in-situ growth, the peak of the Fe/GO membranes shifted to a lower degree due to the insertion of FeO NPs and became lower as the loading increased, e.g., 9.5° for Fe$_{c1.5}$/GO. According to the Bragg's law, the interlayer spaces of GO and Fe$_{c1.5}$/GO were 8.3 and 9.3 Å, respectively. Besides the dry state, the membranes were immersed in water for 1 h to investigate the interlayer space under wetted state. Compared with that under the dry state, the diffraction degrees of the GO and Fe/GO membranes became smaller (Fig. 2j), due to the hydration of nanosheets from the affinity of oxygen-containing groups towards water molecules. For example, the diffraction angles of the wetted GO and Fe$_{c1.5}$/GO membranes decreased to 6.6° and 5.8°, respectively. Under the wetted state, the

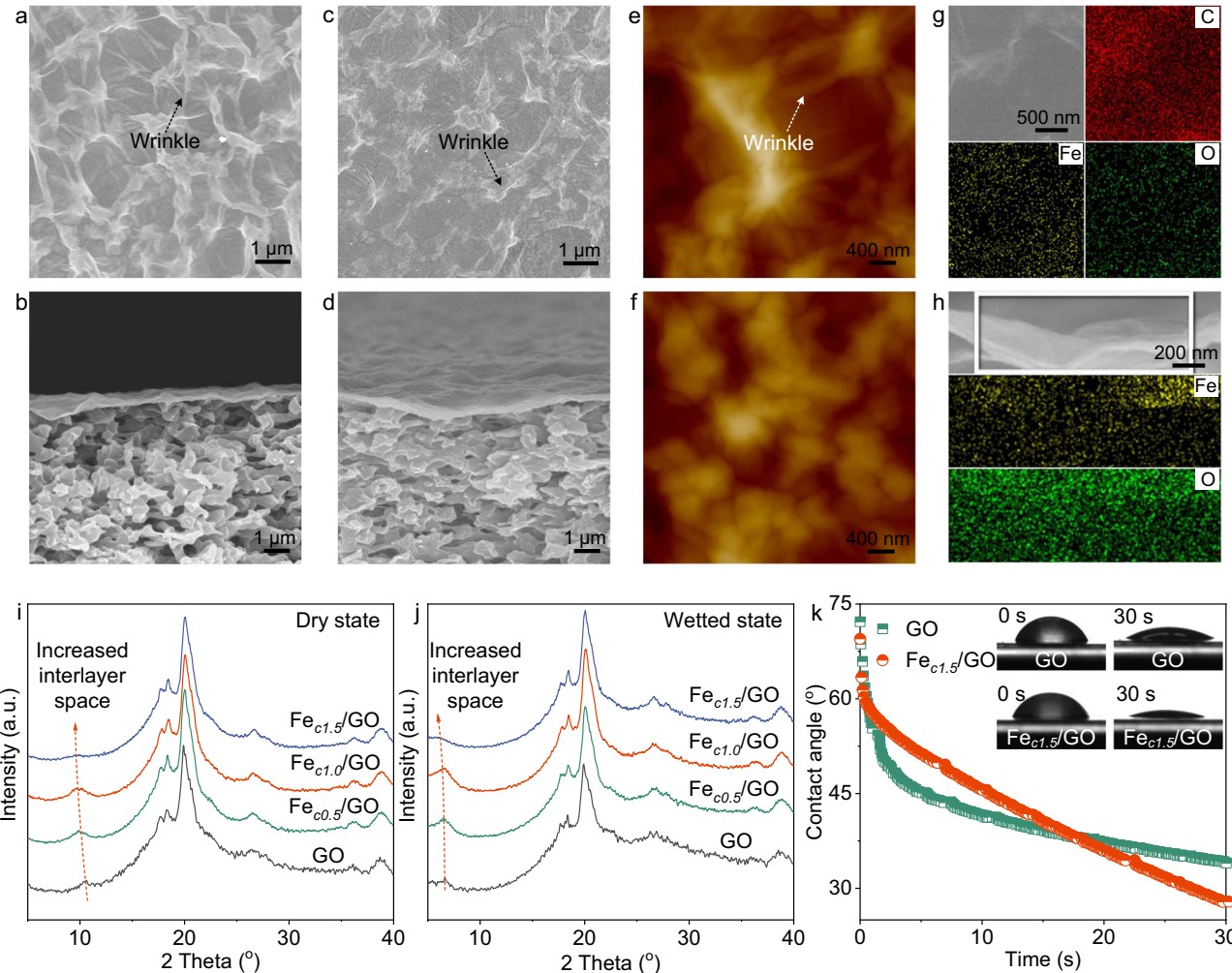

**Fig. 2 | Characterizations of Fe/GO membranes. a, b** Top and cross-sectional view SEM images of the GO membrane. **c, d** Top and cross-sectional view SEM images of the $Fe_{c1.5}$/GO membrane. **e, f** AFM images of the GO and $Fe_{c1.5}$/GO membranes. **g, h** Top and cross-sectional EDS mapping images of the $Fe_{c1.5}$/GO membrane. **i** XRD patterns of the Fe/GO membranes fabricated with different Fe precursor concentrations at dry state. **j** XRD patterns of the Fe/GO membranes fabricated with different Fe precursor concentrations at wetted state. a. u. represents arbitrary unit. **k** Dynamic water contact angle of the GO and $Fe_{c1.5}$/GO membranes.

Fe/GO membranes also had smaller diffraction angles and larger interlayer space than GO, and the interlayer space became wider as FeO loading increased, from 13.3 Å for GO to 15.2 Å for $Fe_{c1.5}$/GO. Apart from the interlayer space, the dynamic water contact angles of the GO and $Fe_{c1.5}$/GO membranes were measured (Fig. 2k). Due to the reduction of GO, the contact angle of $Fe_{c1.5}$/GO was slightly larger than that of GO at the beginning, yet its decline over time was faster as the FeO deposition enhanced the water transport through the $Fe_{c1.5}$/GO layers. Overall, the enhanced hydrophilicity of $Fe_{c1.5}$/GO could remarkably promote the interactions between FeO and aqueous solutions, and thus water dehalogenation.

### Performance of Fe/GO membranes

We evaluated the performance of the membranes for DCAA degradation in the presence of 1.0 mM S(IV). The permeance became larger with increased FeO loading (Fig. 3a), which can be attributed to the increase in interlayer space from the FeO insertion and the decrease of oxygen functional groups from reduction[41]. For example, the $Fe_{c1.5}$/GO membrane exhibited quadrupled permeance compared to that of GO, reaching up to 48.6 L m$^{-2}$ h$^{-1}$ bar$^{-1}$. Interestingly, the reduction efficiency increased simultaneously. For $Fe_{c1.5}$/GO, the efficiency was 96.8%, much > 36.2% for GO. Notably, the Fe/GO membrane fabricated with Fe precursor over 1.5 mmol L$^{-1}$ exhibited poor reduction efficiency

(<10%) due to the existence of defects (Supplementary Fig. 9). Therefore, the $Fe_{c1.5}$/GO membrane was chosen for the subsequent experiments. Figure 3b illustrated that the DCAA reduction was accompanied by the Cl$^-$ formation, indicating that Cl$^-$ was the only Cl-containing product.

We measured the DCAA reduction efficiency under various processes to investigate the roles of S(IV) and FeO. A nearly complete removal of DCAA could be achieved for the $Fe_{c1.5}$/GO membrane in the presence of 1.0 mM S(IV) (Fig. 3c). In contrast, only S(IV) addition exerted little effect on the removal of DCAA, since S(IV) did not generate active radicals via self-decomposition[9]. In the absence of S(IV), the $Fe_{c1.5}$/GO membrane exhibited a removal efficiency of only 35.5 – 50.7%. Since no chlorine-containing product detected in the permeate, this partial removal was likely due to the adsorption in nanochannels. Additionally, the GO membrane without FeO showed a low removal efficiency of 36.2% within 60 min in the presence of 1.0 mM S(IV), suggesting the importance of iron sites in Fe/GO membranes for S(IV) activation and further reduction processes. Furthermore, an additional FeO/GO membrane was synthesized by filtering commercial FeO nanoparticles (20 nm) onto the GO membrane. This composite membrane showed a removal efficiency of 45.6% in the presence of 1.0 mM S(IV) (Fig. 3c). However, the ultrafast DCAA degradation was only observed in Fe/GO membrane catalytic systems,

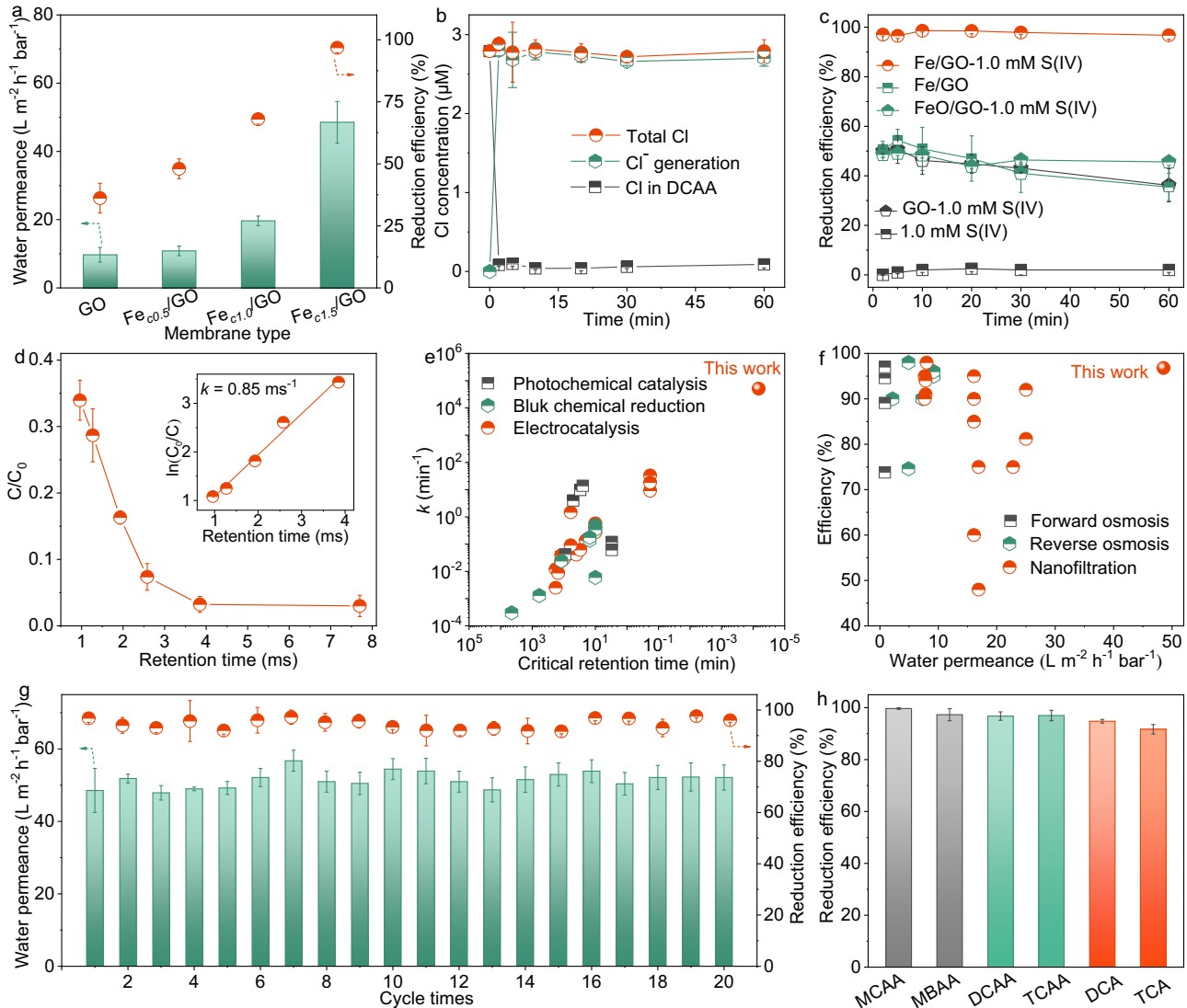

**Fig. 3 | Performance of Fe/GO membranes. a** Water permeance and DCAA reduction efficiency of Fe/GO membranes as a function of iron loading amounts. Conditions of the feed solution: initial DCAA level = 180 μg L⁻¹, initial S(IV) level = 1.0 mM, pH_ini. 7.0 ± 0.1, and 25 ± 0.5 °C. **b** Cl-containing product and the mass balance of the total Cl in Fe_c1.5/GO membranes/S(IV) systems. **c**, Reduction efficiency of DCAA under various processes. Conditions of the feed solution: initial DCAA level = 180 μg L⁻¹, pH_ini. 7.0 ± 0.1, and 25 ± 0.5 °C. **d** Normalized concentration of DCAA (C/C₀) versus membrane retention time. Retention time indicates the ratio of the pore volume within the catalytic membrane over the flow rate through the membrane. As shown in Eq. 1 in the methodology according to a previous study[39], it can be experimentally controlled by adjusting the applied pressure. **e** Comparison of first-order rate constant (k) values. Critical retention time indicates the time for achieving > 90% of removal (Supplementary Table 1). **f** Comparison of permeance and removal efficiency between our Fe/GO membrane and other nanofiltration, forward osmosis, and reverse osmosis membranes (Supplementary Table 2). **g** Stability test of water permeance and reduction efficiency over cycle times. **h** Reduction efficiency of four HAAs and two chlorinated organic pollutants. Conditions of the feed solution: initial level of each HAA = 180 μg L⁻¹, initial levels of DCA and TCA = 0.5 mM, initial S(IV) level = 1.0 mM, pH_ini. 7.0 ± 0.1, and 25 ± 0.5 °C. Error bars represent the standard deviation from at least triplicate experiments. Some of the error bars are smaller than the symbols.

where Fe NPs were loaded in the interlayer spaces of GO, indicating that FeO needs to be placed within a confined space to enhance contact between reactants with FeO, and to decrease energy barriers associated with S(IV) activation and DCAA reduction for achieving unprecedent degradation performance.

Generally, degradation of HAAs is regarded as first-order reactions[42,43]. The calculated first-order rate constant for reducing DCAA by the Fe_c1.5/GO membrane reached 51,000 min⁻¹ (0.85 ms⁻¹) (Fig. 3d), which was six to seven orders of magnitude greater than that by electrocatalytic process (0.0025 – 33.3 min⁻¹), photochemical system (0.042 – 13.9 min⁻¹), and bulk chemical reduction (0.0013 – 0.5 min⁻¹) (Fig. 3e and Supplementary Table 1). Meanwhile, the Fe_c1.5/GO membrane required an extremely short

retention time of a record-breaking 3.9 ms for nearly complete DCAA reduction. By contrast, it would take up to ~600 hrs to achieve the same reduction level in bulk solutions spiked with 50 mg of Fe/GO nanosheets (i.e., 0.5 g L⁻¹) (Supplementary Fig. 10). Besides, in comparison with catalytic systems, the performance of the Fe_c1.5/GO membrane could also easily break the trade-off limitation between water permeability and removal efficiency of membrane separation processes, including nanofiltration, forward osmosis, and reverse osmosis (Fig. 3f and Supplementary Table 2). This high permeability of Fe/GO membranes was attributable to the expanded interlayer space from FeO intercalation and the decreased friction from removal of functional groups[44]. Notably, the Fe_c1.5/GO membrane could achieve >90.8% DCAA

dechlorination for tap water samples with environmentally relevant DCAA concentrations (i.e., 80 µg L$^{-1}$) (Supplementary Fig. 11).

In addition to permeance and degradation efficiency, the stability and universality of the Fe/GO membrane catalytic system were further investigated through pressure-dependent continuous cross-flow experiments. As presented in Fig. 3g, the Fe$_{c1.5}$/GO membrane showed nearly unchanged water permeance and DCAA reduction efficiency over 20 cycle times (with each cycle lasting ~1 h). Meanwhile, the amount of Fe leached during reduction was below the detection threshold of 1.0 µg L$^{-1}$ (Supplementary Fig. 12). Furthermore, the chemical composition of Fe$_{c1.5}$/GO membrane samples before and after cyclic test was assessed by conducting the Fe XPS analysis. Less than 2.3% Fe(II) in the Fe$_{c1.5}$/GO membrane was converted into Fe(III) after operation for 20 hrs (Supplementary Fig. 13), indicating the excellent stability of catalytic sites and the efficient conversion of Fe(II)/Fe(III). Reportedly, Fe(II) exhibits a greater ability for activating S(IV) compared to Fe(III)[16,21–24]; specially, Fe(II) provides electrons to facilitate the formation of Fe(III)-sulfite complexes, and decomposition of these complexes results in the production of Fe(II), with Fe(III) being the immediate that complexes with sulfite. The relatively stable distribution weight of Fe(II) through facilitated electron shuttling likely by the electrically conductive reduced graphene oxide (rGO), further promoted S(IV) activation and DCAA reduction. All above results demonstrated the great stability of the Fe/GO membrane and was attributed to the stably confined FeO NPs within the GO nanosheets. To confirm the generic applicability, the Fe$_{c1.5}$/GO membrane was used to remove a diverse range of pollutants, including haloacetic acids (i.e., MCAA, MBAA, DCAA, and TCAA) and chlorinated organic pollutants (i.e., 1,2-dichloroethane (DCA) and 1,1,1-trichloroethane (TCA)). All the pollutants were rapidly and almost completely reduced (Fig. 3h), with the primary degradation products of halogen ions (i.e., Cl$^-$ and Br$^-$) (Supplementary Fig. 14). Overall, the Fe$_{c1.5}$/GO membrane could perform a robust ARP with great performance, excellent stability, and universal applicability.

**Mechanisms of confinement catalysis and high permeability**

Three main reasons can be utilized to explain ultrahigh reduction efficiency. Firstly, nanoconfinement effects enhance reactants accessibility to catalysts. During the Fe/GO membrane catalysis, to reach the permeate side, all reactants including S(IV) and DCAA had to pass the nanoconfined transport channels between adjacent GO nanosheets, which had been loaded with high-density Fe NPs, consequently largely increasing reactants accessibility to catalysts for accelerating DCAA decomposition. Secondly, nanoconfinement effects increase catalyst-to-reactant ratios (Supplementary Text 1 and Supplementary Table 3). All Fe NPs were loaded in the GO interlayer nanochannels, the reaction regions were nanoconfined as well, thereby substantially increasing catalyst-to-reactant ratios in reaction regions for promoting S(IV) complexation and electron transfer[16,21–24] and thus enhancing S(IV) activation and contaminant degradation. In fact, the effective molar ratios of Fe to sulfite and Fe to DCAA reached as high as 1168–292 and 418963–104741, respectively (see detailed calculation in Supplementary Text 1), which were much higher than those for catalysis in bulk solution (i.e., 0.1–0.5 and 5–20, respectively[15,16,45,46]). Thirdly, nanoconfinement effects greatly reduce energy barrier. Nanoconfinement effects for adjacent GO nanosheets would significantly reduce the energy barrier and facilitate the removal of contaminants, which was demonstrated by density function theory (DFT) simulation as below.

We investigated the energy profile of S(IV) activation within Fe/GO membranes and on FeO NPs using DFT calculations. The activation process of S(IV) falls into three stages: (1) adsorption of S(IV) on FeO surfaces, resulting in the formation of iron(II)-sulfite complex (FeHSO$_3^+$); (2) oxidation of FeHSO$_3^+$ to iron(III)-sulfite complex (FeSO$_3^+$) in the presence of O$_2$; (3) decomposition of FeSO$_3^+$, leading to the formation of sulfite radicals (SO$_3^{\cdot-}$) (Fig. 4a). Amongst these stages,

the generation of SO$_3^{\cdot-}$ through the decomposition of FeSO$_3^+$ was the rate-limiting step. Figure 4a indicated a remarkable decline in the kinetic free-energy barrier for the dissociation of FeSO$_3^+$ into SO$_3^{\cdot-}$ (from 0.41 to 0.86 eV) through Fe/GO, compared to FeO (from -1.44 to 0.86 eV). It further resulted in higher formation of SO$_3^{\cdot-}$ ($\alpha_N = 14.7$, $\alpha_{\beta-H} = 16.0$) for Fe/GO spiked with S(IV) in comparison with FeO/S(IV) (Fig. 4b), implying an important role of spatial confinement of GO matrix towards FeO in S(IV) activation.

We calculated the length of the C-Cl bond for Fe/GO, FeO, GO, and a free environment (Supplementary Fig. 15). Longer length of C-Cl means more easily degradation. The calculated length of the C-Cl bond for Fe/GO and FeO was 1.807 and 1.829 Å, respectively, which was 0.018–0.069 Å longer than the normal C-Cl bond for GO membranes (1.789 Å) and a free DCAA molecule (1.760 Å), suggesting that iron active sites can bind DCAA and intermediates. From the energy profiles associated with the degradation of DCAA by Fe/GO and FeO in the presence of S(IV), it was obvious that the conversion of CHCl$_2$COOH to CH$_2$ClCOOH was the rate-limiting step (Fig. 4c). Compared to FeO (−0.49 eV to 2.02 eV), Fe/GO required the lower kinetic free-energy barrier associated with the transformation of CHCl$_2$COOH into CH$_2$ClCOOH, from −0.57 eV to 1.67 eV, in the presence of S(IV). These processes would undergo a remarkable amplification by a combination of the spatial confinement during the activation of S(IV). These results indicated that confinement effects remarkably reduced the energy for the rate-limiting steps, i.e., FeSO$_3^+$ dissociation into sulfite radicals in activation process and CHCl$_2$COOH transformation into CH$_2$ClCOOH in degradation process, together favoring the reduction of DCAA and reaction intermediates on the iron centers through Fe/GO membranes.

Apart from great reduction efficiency, for water permeation of GO membranes, the molecules pass through defects/edges of nanosheets and interlayer spaces between adjacent nanosheets[47,48]. Due to the two-dimensional configuration, the interlayer channels primarily govern the permeance of GO membranes. We calculated the diffusion processes of water molecules through the interlayer space of GO and Fe/GO membranes using molecular dynamic (MD) simulations. Results in Fig. 4d, e showed that the diffusion constant for the Fe/GO membrane was $1.58 \times 10^{-5}$ cm$^2$ s$^{-1}$, larger than $1.21 \times 10^{-5}$ cm$^2$ s$^{-1}$ for GO. This difference could be attributed to the expanded interlayer space from 13.3 Å for GO to 15.2 Å for Fe$_{c1.5}$/GO. Meanwhile, the higher density of water on GO than on Fe/GO from the more oxygen-containing groups implied the greater interaction, thereby reducing water diffusion process (Supplementary Fig. 16). In other words, the friction experienced by water molecules within the interlayer spaces of Fe/GO membranes was lower than that within GO. However, the diffusion constant ratio of Fe/GO to GO was 123%, lower than the quadruple permeance ratio observed in experimental results. Although molecular transports were heavily affected by interlayer channels, the free spaces within GO membranes, resulting from irregular stacking and nanoparticle insertion, could provide transport channels and reduce tortuosity to enhance water permeance[49]. Benefiting from the insertion of FeO NPs, a considerable number of free spaces would be formed, thereby further improving water permeance.

## Discussion

We have constructed the Fe/GO membranes with uniform FeO NPs and proved their great performance for dechlorination. Because of enhanced reactants accessibility to catalytic sites, increased catalyst-to-reactant ratios in reaction regions, and promoted activation of FeSO$_3^+$ to sulfite radical and conversion of DCAA to MCAA, the membranes show great reduction efficiency for degradation of DCAA to chloride, with a first-order rate constant of 51,000 min$^{-1}$, which is six to seven orders of magnitude greater than conventional catalytic systems. Because of the adjustment from the growth of FeO NPs to membrane structure, chemical composition, and interlayer space, the

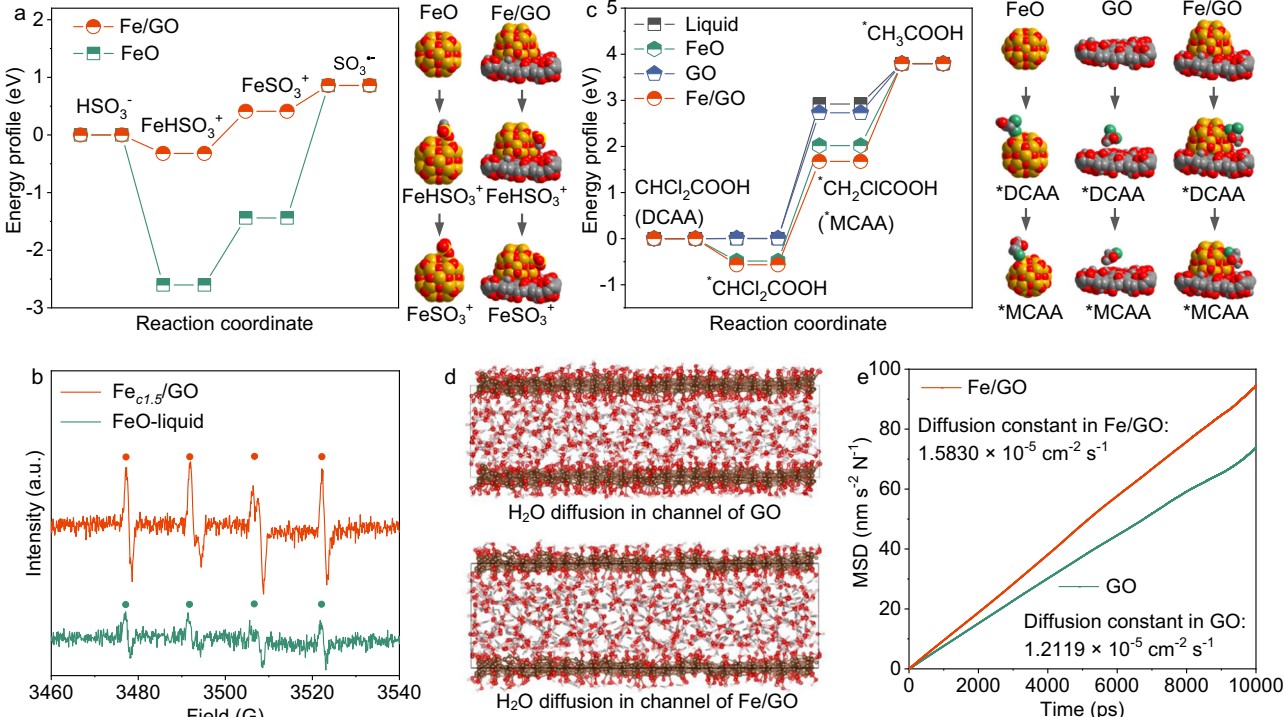

**Fig. 4 | Calculated reaction mechanism of confinement catalysis and superior permeability. a** Energy profiles of S(IV) dissociation through Fe/GO and FeO. Chemical structures of FeO, $FeHSO_3^+$, and $FeSO_3^+$ are presented in the first column, while Fe/GO, $FeHSO_3^+$, and $FeSO_3^+$ are displayed in the second column. **b** EPR spectra of the 5,5-dimethyl-1-pyrrolidine-N-oxide (DMPO) spin-trapping for $Fe_{c1.5}$/GO membrane catalytic systems and FeO/S(IV) bulk solutions. Circle symbols indicate signals of the $DMPO/SO_3^{\cdot-}$ adduct. a. u. represents arbitrary unit. Conditions: DMPO concentration = 100 mM, initial S(IV) level = 1.0 mM, FeO

dosage = 1.0 g $L^{-1}$, $pH_{ini.}$ 7.0 ± 0.1, and 25 ± 0.5 °C. **c** Energy profiles of DCAA degradation in Fe/GO, FeO, GO, and a free environment in the presence of S(IV). In the first column, the structures are presented for FeO, DCAA, and MCAA bound to FeO. In the second column, the structures are displayed for GO, DCAA, and MCAA bound to rGO. In the third column, the structures of Fe/GO, DCAA, and MCAA bound to Fe/GO are presented. **d**, **e** Diffusion of $H_2O$ in interlayer channels of GO and Fe/GO. The orange line and green line represent Fe/GO and GO, respectively.

membranes exhibit a permeance of 48.6 L $m^{-2}$ $h^{-1}$ $bar^{-1}$, which is quadrupled compared to that of GO. In addition, the Fe/GO membranes display excellent stability and maintain their performance over 20 cycles. We have further demonstrated that the membranes are applicable for environmentally relevant concentrations and for a wide range of halogenated organic pollutants. Although the efficiency may be further improved by tuning chemical compositions, controlling nanoparticles and membrane structures, and applying other types of nanosheets and nanoparticles, the concept of designing catalytic membranes based on nanoparticles and two-dimensional nanosheets paves the way for enabling strong nanoconfinement effects for efficient dehalogenation and water treatment.

## Methods
### Materials and chemicals
MCAA (anhydrous, 99.0%), MBAA (anhydrous, ≥99.0%), DCAA (anhydrous, ≥99.0%), TCAA (anhydrous, ≥99.0%), DCA (anhydrous, 99.8%), TCA (97%), and $FeCl_3$ (anhydrous, 98.0%) were purchased from Sigma-Adrich (USA). Graphite powder (99.0%), sodium sulfite ($Na_2SO_3$, ≥98.0%), sodium sulfate ($Na_2SO_4$, ≥99.0%), and sodium borohydride ($NaBH_4$, 98.0%) were purchased from Sinopharm Chemical Reagent Co. Ltd. (Shanghai, China). The 5,5-dimethyl−1-pyrrolidine-N-oxide (DMPO, ≥97.0 wt%) was obtained from Aladdin Chemistry Co. Ltd. (Shanghai, China). All chemicals were at least of analytical grade and used as received. Hydrophilic polyvinylidene difluoride membrane with 0.22 μm pore size was obtained from Merck (New Jersey, USA). All solutions were prepared in Milli-Q ultrapure water (18.2 MΩ cm, Millipore).

### Fabrication of Fe/GO membranes
GO was prepared by oxidizing natural graphite powder using the Hummers method[50–52]. Graphite powder (2.0 g) and $NaNO_3$ (1.0 g) were added in concentrated $H_2SO_4$ (48 mL) and the temperature was controlled using an ice bath. Then, $KMnO_4$ (6.0 g) was slowly added under stirring to prevent the temperature from exceeding 20 °C. After reaction for 2 hrs, the suspension was thermally treated at 35 °C for 1 h. Water (92 mL) was slowly added in the suspension, following by heat-treatment at 98 °C for 40 min and addition of 30% $H_2O_2$ solution. Ultimately, the product was washed with dilute HCl solution, collected, and dried. Colloidal dispersions of individual GO sheets in water (150 mL, 20 mg $L^{-1}$) were prepared using an ultrasonic homogenizer (JY98-IIIDN, Scientz) with ultrasound treatment at 40% power for 2 hrs. A solution of ferric iron ions (20 mL, 0.5–1.5 mmol $L^{-1}$) was prepared and added to the GO dispersion (150 mL, 20 mg $L^{-1}$). After adsorption for 10 hrs, the mixture was reduced by adding $NaBH_4$ in a controlled amount (i.e., 1.6 M, 75 μL). Subsequently, Fe/GO membranes were fabricated by vacuum filtration of the mixed solution (20.0 mL) under a vacuum pressure (0.95 bar). The membranes were then dried overnight at room temperature before further application. The resulting Fe/GO membranes were cut into rectangular strips of about 30 mm × 100 mm for testing, without any additional modification. Notably, excessively thin membranes (i.e., 80 μg GO) resulted in reduced removal efficiency, while excessively thick membranes (i.e., 1600 μg GO) led to decreased water permeance (Supplementary Fig. 17). Therefore, the membranes with moderate thickness (i.e., 400 μg GO) that was loaded with Fe NPs were selected in our study to achieve both high water permeance and excellent removal efficiency.

## Characterization

The morphology of Fe/GO membranes was characterized using a STEM (Talos F200X STEM, Thermo Scientific) operating at 200 kV. To capture the TEM images of nanosheets, a TEM (JEM-2100, JEOL Ltd.) with an accelerating voltage of 200 kV was employed. For sample preparation, the GO or Fe/GO suspension was drop-cast onto a copper mesh-supported carbon film and dried at room temperature. The morphology and elemental distribution of the fabricated membranes were observed using a field-emission scanning electron microscope (SEM, Hitachi S-4800, Germany) with an accelerating voltage of 5 kV, equipped with an energy-dispersive spectrometer. To minimize the recharging effect and facilitate membrane sample preparation, the membrane was fractured in liquid nitrogen and subsequently coated with an ultrathin platinum layer. To investigate the nanosheet configuration and structure, an AFM (Multimode nanoscope, Bruker, USA) was utilized. NanoScope Analysis software was used to analyze images and height profiles. For sample preparation, the GO or Fe/GO suspension was drop-cast onto a mica plate and dried at room temperature. Furthermore, an XPS was carried out using an RBD (RBD Enterprises, USA) upgraded PHI-5000C ESCA system (Perkin Elmer) with monochromatic Mg Kα X-rays ($h\nu = 1253.6$ eV) at 250 W. To ensure sufficient sensitivity and resolution, the high voltage was maintained at 14.0 kV, the pass energy was set to 46.95 eV, and the pressure in the analysis chamber was kept below $5 \times 10^{-8}$ Pa. To measure the interlayer space of the membranes, an XRD (D8 Advance, Bruker Co.) with Cu Kα radiation ($\lambda = 0.154056$ nm) was used in a continuous scanning mode. For dry samples, the membrane was dried at 50 °C before measurement. For wetted samples, the membrane was immersed in water for 1 h before measurement. The interlayer space was calculated using Bragg's Law of $2d\sin\theta = n\lambda$, where d and θ were the interlayer space and characteristic peak theta angle, respectively; and λ and n were 0.154056 nm and 1, respectively. Raman spectra were collected using a WITEC micro-Raman at a 532 nm laser. The dynamic contact angle of the prepared membranes was recorded using an optical contact angle and interface tension meter (ST200KB, USA KINO Industry Co.).

Concentrations of haloacetic acids (i.e., MCAA, MBAA, DCAA, and TCAA) and their products (i.e., Cl⁻ and Br⁻) were quantified by an ion chromatograph (Dionex Aquion RFIC). A Dionex AS19 analytical column (4 × 250 mm) and an AG19 guard column (4 × 50 mm) were applied to separate them. Potassium hydroxide (20 mM) was used as effluents. Also, the measurement was carried out at a flow rate of 1.0 mL min⁻¹, a constant suppressor current of 50 mA, column temperature at 30 °C, and a duration of ~20 min. Prior to ion chromatography analysis, all the samples were filtered using a 0.22 μm filter (Nylon, Titan). Concentrations of chlorinated organic pollutants (i.e., DCA and TCA) were analyzed by gas chromatograph using an electron capture detector (6820, Agilent). The analysis was performed under the column temperature of 100 °C and hydrogen flow rate of 1.5 mL min⁻¹. Elemental analyses, including iron concentrations, were performed by inductively coupled plasma mass spectrometry (ICP-MS, Agilent 7500 s). The detention limit for Fe was 1.0 μg L⁻¹. Solution pH was determined at room temperature using a pH meter (MM374, Hach) calibrated with standard buffers routinely at pH 4.01, 7.01, and 10.01 prior to measurements. Moreover, electron paramagnetic resonance (EPR) experiments were carried out by using a JEOL FA200 EPR spectrometer with DMPO as a spin-trapping agent. All the EPR spectra were obtained at room temperature with a center field of 3350 Gs, a sweep width of 100 Gs, a 9.4 GHz microwave with a power of 1.0 mW, a modulation amplitude of 0.3 Gs, and a sweep time of 41.96 s. The concentration of DMPO, sulfite, and FeO as well as solution pH were 100 mM, 1.0 mM, 1.0 g L⁻¹, and 7.0 ± 0.1, respectively, in all EPR analyses. The hyperfine splitting constants (i.e., $\alpha_N$ and $\alpha_{\beta\text{-}H}$) related to peak distances were utilized to identify radicals.

## Oxyanion reduction experiments

The performance of Fe/GO membranes for DCAA reduction was conducted using a laboratory-scale cross-flow membrane test system. The schematic of the filtration system with all components including recirculation is shown in Supplementary Fig. 18 of the Supporting Information[53,54]. Each membrane coupon was installed in a cross-flow cell (CF042, Sterlitech) with an effective area of 8.0 cm². During catalytic filtration experiments, the stock solution containing sulfite (i.e., 1 mM S(IV)) and haloacetic acids (i.e., 80 or 180 μg L⁻¹ MCAA, MBAA, DCAA, and TCAA) or chlorinated organic pollutants (i.e., 0.5 mM DCA and TCA) was prepared, and then it has been utilized as the feed to the reaction membrane module under a flow rate of about 0.2 L min⁻¹, a pressure of 1.0–2.0 bar, and 90% of power. Unless specified otherwise, all the catalytical filtration tests were performed with only the retentate recirculated back to the feed tank, as presented in Supplementary Fig. 18. To ensure this recirculation does not affect the oxyanion removal performance, additional catalytical filtration tests without recirculation were also included for comparison in Supplementary Fig. 19. For the long-term performance testing of Fe/GO membrane catalytic systems, the feed solution, including S(IV) and haloacetic acids, was replenished with a solution volume of 50–100 mL at regular intervals of 1.0 h between cycles; each cycle represents at least 50–100 mL of clean permeate water obtained by membrane catalytic systems. After the catalytic filtration experiment, the permeate was collected at predetermined sampling time (i.e., 2, 5, 10, 20, 30, and 60 min), and immediately analyzed for halogenated organics reduction, product formation, and iron leaching. All the experiments were carried out at room temperature. It is worthwhile to note that the shortest sampling time was 2 min in the current study due to the time needed for the permeate water to reach to the sampling point.

In addition, in suspension systems containing $Fe_{c1.5}$/GO composites/S(IV), reduction reactions were initiated by introducing DCAA, $Fe_{c1.5}$/GO composites, and S(IV) into the solutions. Stock solutions of Na₂SO₃ were freshly prepared prior to each set of experiments. Samples were withdrawn at predetermined time of 0, 0.5, 1, 2, 3, 4, 6, and 8 hrs, and the residual S(IV) in the samples was quenched by adding H₂O₂.

## Reaction kinetics model

The retention time can be calculated by Eq. 1:

$$t = \frac{V}{Q} = \frac{S \times h}{J \times S \times P} = \frac{h}{J \times P} \tag{1}$$

Where $V$ is the pore volume of membrane (cm³), $Q$ is the flow rate (mL min⁻¹), $S$ (m²) is the effective membrane filtration area, h (nm) is the effective thickness of interlayer free space, and $J$ (L m⁻² h⁻¹ bar⁻¹) and $P$ (bar) are water permeance and applied pressure, respectively.

Additionally, the observed rate constants ($k$) for DCAA reduction were calculated based on the pseudo-first-order kinetics model shown in Eq. 2, where $r$ is the rate of DCAA reduction (μg min⁻¹), $C_{DCAA}$ represents DCAA concentration (μg), and $t$ indicates retention time, i.e., the hydraulic residence time within the catalytic membrane (min).

$$r = \frac{-\mathrm{d}C_{DCAA}}{\mathrm{d}t} = kC_{DCAA} \tag{2}$$

## Computational methods

Molecular dynamics simulations were performed using the GROMACS software package (version 2021.3)[55–57]. The systems containing GO and rGO with different C:O ratios and layer distances were constructed using the gmxtools software[58], and the SPC water model was filled into the GO and rGO systems[59]. The atomic interactions in GO and rGO were parameterized using the general AMBER force field (GAFF)[60]. After

energy minimization, production runs were carried out in the NVT ensemble at 300 K with a time step of 1 fs. The system temperature was controlled using a v-rescale thermostat ($\tau_T = 1\,ps$). After 10 ns of simulation, the mean squared displacement (MSD) of the water molecules was analyzed by using the GROMACS toolkits.

Moreover, we calculated the free energy diagram of the conversion from $HSO_3^-$ to $SO_3^{\cdot-}$ and from DCAA to CAA in Fe/GO, FeO, rGO, and a free environment in the presence of S(IV) by using the CP2K package (version 2023) in the framework of the density functional theory[61], according to the hybrid Gaussian and plan-wave scheme[62]. The molecular orbitals of the valence electrons were expanded into DZVP-MOLOPT-SR-GTH basis sets[63], while atomic core electrons were described using Goedecker-Teter-Hutter (GTH) pseudopotentials[64]. A plane-wave density cutoff of 500 Ry was adopted. The long-range van der Waals interactions have been described by the DFT-D3 approach[23]. All the structures were fully relaxed using the BFGS scheme in CP2K, and the force convergence criterion was set to $4.5 \times 10^{-4}$ hartree bhor$^{-1}$. In the calculations of FeO in a liquid environment, the energies were calibrated using the SCCS continuum solvation model[65–67].

### Statistical analysis
Experiments were repeated at least three times, and error bars represent ± one standard deviation from the mean of the independent triplicates. Student's $t$-test (two-tailed, unpaired, assuming equal variance) was used to determine if differences in oxyanion removal kinetics were significant (e.g., $p < 0.05$).

## Data availability
The data supporting the findings of this study are included in the main text and supplementary information filed. Additional data are available from the corresponding author upon request. Source data are provided with this paper.

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

## Acknowledgements

This work was substantially supported by NSFC-RGC Funding Scheme (N_HKU721/22 to C.T.) from the Research Grants Council of the Hong Kong Special Administration Region, China. The research was also sponsored by the Hong Kong Scholar Program (No. XJ2021035 to Q.X.), and the Shanghai Pujiang Program (No. 22PJD075 to Q.X.). We appreciate the staff from the Electron Microscope Unit of The University of Hong Kong for SEM and TEM sample preparation and analysis. We thank the high-performance public computing service platform of Jinan University for providing computational resources.

## Author contributions

Q.X., W.L., and C.T. conceived the idea and designed the research. Q.X. S.X., and L.W. performed the research. Q.X., W.L., and C.T. interpreted the data and wrote the manuscript.

## Competing interests

The authors declare no competing interests.
