## [Transparent Peer Review file · Nature Communications]

Ultrafast complete dechlorination enabled by ferrous oxide/graphene oxide catalytic membranes via nanoconfinement advanced reduction

Corresponding Author: Professor Chuyang Tang

Version 0:

Reviewer comments:

Reviewer #1

(Remarks to the Author)

The topic of this study is interesting. The authors fabricated a ferrous oxide/graphene oxide (Fe/GO) catalytic membrane and observed its ability to rapidly and completely dechlorinate dichloroacetic acid (DCAA) using sulfite radicals, achieving nearly 100% reduction efficiency and a remarkably high rate constant. While the results are impressive, the lack of detailed experimental information hinders the strength of the conclusions.

1. How about the contribution of the PVDF membrane in the removal of DCAA?
2. The quantity of materials used in each membrane piece, as per the "Fabrication of the Fe/GO membranes" section, seems limited, with approximately 0.4 mg of GO (20 mg/L * 20 mL) and 1-2 mg of FeO (1.5 mmol/L * 20 mL = 3 uM ferrous ion, the ultimate amount of FeO was dependent on the dose of NaBH₄). Is this amount adequate to form a membrane of sufficient thickness? The membrane thickness might be an important parameter in determining the performance of the catalytic membrane.
3. The rationale behind using a 1 mM sulfite concentration should be explained. What is the expected stoichiometric ratio of Fe to sulfite in this context?
4. The introduction section emphasizes the need for DCAA dechlorination. However, the focus should shift towards discussing how the combination of FeO and sulfite enhances the catalytic membrane's efficiency. Therefore, the reaction mechanism between FeO and sulfite should be addressed in the introduction to justify the experimental design.
5. The methodology of the stability test is unclear. Were there any treatments between cycles?
6. The data in Supplementary Table 1 lack detail. The first-order rate constant is influenced by various experimental conditions, such as the initial concentrations of HAAs. Please provide the concentrations of HAAs and the material dosages used, if applicable.
7. Supplementary Table 2 compares the permeance and removal efficiency of the Fe/GO membrane with other nanofiltration and reverse osmosis membranes. How is the process of the Fe/GO membrane defined? Is it a nanofiltration membrane?
8. The clarity of the figures is crucial for understanding the results. In Fig. 3a, where two sets of data are presented (water permeance and DCAA reduction efficiency), it is important to clarify the correspondence between the columns/scatter plots and the two parameters to aid interpretation. Similar issue exists in several other figures.
9. Regarding the experiments for Figure 3d, it is essential to provide details on how they were conducted. Specifically, what methodology was employed to determine the concentration of DCAA within 1 ms? This information is critical for understanding the experimental setup and the rapid kinetics involved in the dechlorination process.

Reviewer #2

(Remarks to the Author)

The present manuscript reported Ultrafast complete dechlorination enabled by ferrous oxide/graphene oxide catalytic membranes via nanoconfinement advanced reduction. The authors constructed Catalytic membranes by graphene oxide nanosheets with integrated ultrafine and monodisperse sub-5 nm nanoparticles, and found The catalytic membrane achieves ultrafast and complete dechlorination of 180 µg L⁻¹ dichloroacetic acid to chloride enabled by sulfite radicals, with nearly 100% reduction efficiency within a record-breaking 3.9 ms, accompanied by a six to seven orders of magnitude

greater first-order rate constant of 51,000 min⁻¹ than current catalytic processes. This sounds to be interesting. However, I could not recommend it to be accepted for publication by this journal because its novelty is not well documented. Some of my concerns are given below.

- 1) The novelty is not well documented. The apparent catalytic performance of the FeO/GO membrane is much higher than the FeO/GO dispersions. The reasons for this great difference are not well clarified. It is necessary to use the nanoconfinement effect to quantitatively explain the huge difference. In both the abstract, and the introduction, it is not possible to understand the driving force for the catalytic dechlorination.
- 2) There are some mistakes in Figure 1i, and the Fe XPS spectra should be re-analyzed. Why FeO NPs were obtained by adding FeCl₃ solution to the GO suspension? The distribution weights of Fe(II) and Fe(III) should be carefully explained. It is also necessary to explain the contributions of Fe(II) and Fe(III) components to the catalytic activity.
- 3) Pages 3-4: "membranes with higher FeO loadings appeared darker due to the GO reduction": Why was GO reduced?
- 4) Pages 3-4, "which might be beneficial to immobilize the mass transfer channels during filtration under pressure and then improve permeation properties.": These channels are substantially not in the direction of the pressure flow of water.
- 5) The use of S(IV) with initial level of 1.0 mM should be given as early as possible.
- 6) The system of Fe-oxide NPs + GO + S(IV) should be added as a control in Fig. 3.
- 7) Fig. 3e summarized the first-order rate constant for reducing DCAA, but the reaction conditions should be listed in detail such as catalysts dosage, S(IV) concentration and so on.
- 8) Is there other radicals in the system except the SO₃^{•-}, such as active hydrogen atoms (H^{*}) or electrons.
- 9) The ultra-fast dechlorination of DCAA on Fe/GO membrane was considered to be attributed to the nanoconfinement effects from GO nanosheets from your DFT calculations. Why the dechlorination efficiency is not well by using Fe/GO nanosheets as catalysts in bulk solutions (Supplementary Figure 10)? Why the catalytic activity could increase orders of magnitudes on the Fe/GO membrane?

Version 1:

Reviewer comments:

Reviewer #1

(Remarks to the Author)

Most of the review comments have not been adequately addressed. The experimental remain insufficient, and the calculation of the material ratio unclear. Without these critical data, it is challenging to evaluate the performance of the membrane and, consequently, value of this work.

1. P3, Line 92: "We synthesized the Fe/GO composite nanosheets through in-situ growth of FeO nanoparticles by introducing a 1.5 mmol L⁻¹ FeCl₃ solution to the GO suspension and reducing Fe³⁺ with NaBH₄." What are the concentration and volume of the GO suspension? Were all of the dosed Fe³⁺ completely transformed into FeO and Fe₂O₃? Was any zero-valent iron generated during the reduction with NaBH₄?
2. Supplementary Fig. 17: The x-axis contains three mass data points. Are these the masses of the Fe/GO membranes? The calculation missed some important parameters, such as how many volumes of the feed solution was treated in one hour.
3. "the molar ratio of Fe to sulfite in reaction regions of Fe/GO membranes was ultrahigh" "the molar ratios of Fe to sulfite and Fe to DCAA were calculated as 0.000234 and 0.168." What do you mean by using the word "ultrahigh"? What is the mechanism in the Fe/GO mediated catalysis?
4. Without the aforementioned experimental details, the conclusions drawn in this work are not well supported.

Reviewer #2

(Remarks to the Author)

The comments have been well responded, and the present version of this manuscript is possible accepted for publication.

Version 2:

Reviewer comments:

Reviewer #1

(Remarks to the Author)

I have no further comments. The manuscript is likely acceptable for publication.

Response to the comments from the Reviewers

Reviewer #1 (Remarks to the Author):

Comments:

The topic of this study is interesting. The authors fabricated a ferrous oxide/graphene oxide (Fe/GO) catalytic membrane and observed its ability to rapidly and completely dechlorinate dichloroacetic acid (DCAA) using sulfite radicals, achieving nearly 100% reduction efficiency and a remarkably high rate constant. While the results are impressive, the lack of detailed experimental information hinders the strength of the conclusions.

Response/Action:

We appreciate the reviewer's high evaluation of our study. We have revised the manuscript thoroughly with additional experimental data to better support our findings. Furthermore, detailed point-to-point responses to the reviewer's comments have been prepared below.

1. How about the contribution of the PVDF membrane in the removal of DCAA?

Response/Action:

We thank the reviewer for his/her comments. It is believed that the PVDF membrane did not contribute to the removal of DCAA. There may have two ways for DCAA removal by membrane, i.e., rejection and adsorption. For rejection, the PVDF membrane is microfiltration membrane with a large pore size of 0.22 μm and has no rejection for DCAA. For adsorption, the PVDF membrane usually has poor adsorption capacity for contaminant adsorption. We have used the PVDF

membrane to remove DCAA; the removal efficiency for DCAA was limited (<2%), demonstrating that the PVDF membrane has not contributed to DCAA removal.

2. The quantity of materials used in each membrane piece, as per the "Fabrication of the Fe/GO membranes" section, seems limited, with approximately 0.4 mg of GO ($20 \text{ mg/L} * 20 \text{ mL}$) and 1-2 mg of FeO ($1.5 \text{ mmol/L} * 20 \text{ mL} = 3 \text{ uM}$ ferrous ion, the ultimate amount of FeO was dependent on the dose of NaBH_4). Is this amount adequate to form a membrane of sufficient thickness? The membrane thickness might be an important parameter in determining the performance of the catalytic membrane.

Response/Action:

Following the reviewer's comments, the effect of the Fe/GO membrane thickness on the water permeance and DCAA removal efficiency has been studied. Results were shown in new Supplementary Fig. 17. It was observed that excessively thin membranes resulted in the decline of removal efficiency, while excessively thick membranes led to decreased water permeance. Comparatively, the membrane with the moderate thickness in our study yielded both high water permeance and excellent removal efficiency.

We have revised the manuscript (Lines 350–354). *“Notably, excessively thin membranes (i.e., 4 mL of 20 mg L^{-1} Fe/GO) resulted in reduced removal efficiency, while excessively thick membranes (i.e., 80 mL of 20 mg L^{-1} Fe/GO) led to decreased water permeance (Supplementary Fig. 17). Therefore, the membranes with moderate thickness by introducing 20 mL of 20 mg L^{-1}*

Fe/GO were selected in our study to achieve both high water permeance and excellent removal efficiency.”

“Supplementary Fig. 17. Water permeance and DCAA reduction efficiency of Fe/GO membranes as a function of Fe/GO loading amounts. Conditions of the feed solution: initial DCAA level = 180 μg L⁻¹, initial S(IV) level = 1.0 mM, pH_{ini} 7.0 ± 0.1, and 25 ± 0.5 °C. Error bars represent the standard deviation from at least triplicate experiments.”

3. The rationale behind using a 1 mM sulfite concentration should be explained. What is the expected stoichiometric ratio of Fe to sulfite in this context?

Response/Action:

We agree with the reviewer that the molar ratios are important for the removal performance of catalytic membranes. For catalysis in bulk solution, greater sulfite concentrations generally lead

to faster pollutant removal (*Chem Eng J* **417**, 129115 (2021); *J Hazard Mater* **418**, 125940 (2021)). Considering the maximum concentration of 250 mg L⁻¹ for sulfate regulated by the Chinese National Standards for Drinking Water Quality (GB5749-2022), 1.0 mM sulfite was adopted in this study. Additionally, the molar ratio of Fe to sulfite in reaction regions of Fe/GO membranes was ultrahigh (i.e., 1168–292, see detailed calculation in Supplementary Text 1) compared to that of the bulk solution based catalyst (i.e., 0.1–0.5 (*Chem Eng J* **417**, 129115 (2021); *J Hazard Mater* **418**, 125940 (2021); *J Hazard Mater* **384**, 121497 (2020))). It should be noted that the processes of catalysis in bulk solution and catalysis by membranes are different.

“Supplementary Text 1. Calculation of molar ratios of Fe to sulfite and Fe to DCAA.

In this study, the Fe_{c1.5}/GO membrane had C, O, and Fe atomic contents of 75.98%, 23.27, and 0.75%, respectively. The Fe mass ratio of Fe_{c1.5}/GO was 0.0326 g g⁻¹. For membrane catalytic systems, including DCAA/sulfite solution and Fe_{c1.5}/GO membrane, the molar ratios of Fe to sulfite and Fe to DCAA were calculated as 0.000234 and 0.168, respectively. However, for membrane catalysis, the catalyst and reactants are confined in the transport channels of membranes. Therefore, the molar ratios in reaction regions in membranes are totally different from those in the bulk systems. As demonstrated in previous study¹, the GO membranes could adsorb water and salt solutions with ratios of 1–4 g g⁻¹. Considering the sulfite concentration of 1 mM and the sulfite rejection of 50% through the Fe_{c1.5}/GO membrane, as well as the DCAA concentration of 180 μg L⁻¹ and the limited rejection of membrane for DCAA (<2%), we could calculate that the sulfite and DCAA mass ratios to GO-adsorbed water were 4.1×10⁻⁵ and 1.8×10⁻⁷ g g⁻¹, respectively. Combining the two contents of GO-adsorbed solution and GO-loaded Fe, the molar ratios of Fe to sulfite and Fe to DCAA in the Fe_{c1.5}/GO membrane were estimated to be 1168–292 and 418960–

104740, respectively. The ultrahigh ratios of Fe to sulfite, Fe to DCAA, and sulfite to DCAA at confined interlayer spaces of membranes are beneficial to enhanced catalytic performance.”

4. The introduction section emphasizes the need for DCAA dechlorination. However, the focus should shift towards discussing how the combination of FeO and sulfite enhances the catalytic membrane's efficiency. Therefore, the reaction mechanism between FeO and sulfite should be addressed in the introduction to justify the experimental design.

Response/Action:

The reviewer has a good question. In this study, “*Catalytic membranes can integrate dual functions of filtration and catalysis*^{28–31}. Membranes can harness catalysts and spatially confine reactants and active sites at the micrometer- and nanometer-scale, which offer the potential to enhance catalytic performance; meanwhile, catalysis can facilitate decomposition of contaminants that cannot be effectively rejected by membranes. (Lines 51–54)” Following the reviewer’s suggestion, the reaction mechanism between FeO and sulfite and the discussion about the combination of catalysis and membrane separation processes have been included in the Introduction Section.

Lines 46–48 “*The ARPs generate reactive reducing species, such as sulfite radicals, and facilitate the cleavage of carbon-halogen bonds (i.e., C-Cl)*^{19,20}. For catalysis, however, the decomposition rate ($\sim 0.15\text{ s}^{-1}$) of transition metal ions-sulfite complexes was relatively slower^{21–}

Lines 61–69 “*We, thus, posit designing ultrafine iron-based NP/graphene oxide composite membranes for advanced reduction of HAAs, which provide nanoconfinement effects to achieve reduced activation energy for the formation of reducing radicals for DCAA dechlorination. Furthermore, the reductants, including S(IV) and contaminants, must undergo nanoconfined transport through the catalyst-loaded interlayer channels of the iron-based NP/graphene oxide composite membranes. This greatly enhances the reductant-catalyst contact probability compared to the bulk solution based catalysts. Additionally, the catalysts and reactions will be nanoconfined in the interlayer transport nanochannels of GO membranes, which will substantially increase the catalyst-to-reactant ratios in confined reaction regions and then promote contaminant degradation.*”

5. The methodology of the stability test is unclear. Were there any treatments between cycles?

Response/Action:

There was no treatment for the membranes between cycles. To further address the reviewer’s comment, we have described the methodology of the stability test in detail as follows.

Lines 410–413 “*For the long-term performance testing of Fe/GO membrane catalytic systems, the feed solution, including S(IV) and haloacetic acids, was replenished with a solution volume of 50–100 mL at regular intervals of 1.0 hour between cycles; each cycle represents at least 50–100 mL of clean permeate water obtained by membrane catalytic systems.*”

6. The data in Supplementary Table 1 lack detail. The first-order rate constant is influenced by various experimental conditions, such as the initial concentrations of HAAs. Please provide the concentrations of HAAs and the material dosages used, if applicable.

Response/Action:

To address the reviewer’s comment, we have provided additional experimental conditions, such as the initial concentration of HAAs, types of catalysts, and the catalyst dosage, in the Table 1 of the revised Supporting Information.

Supplementary Table 1. First-order rate constants and corresponding experimental conditions for HAAs removal in this study and previous studies.

No.	Processes	Types of HAAs	Conc. ($\mu\text{g L}^{-1}$)	Reaction time (min)	Rate (min^{-1})	Remarks	Ref.
1	Chemical reduction in bulk solutions	BDCAA	29,120	15	0.177	13.9 g L ⁻¹ Fe ⁰	2
2		CDBAA	20,800	120	0.024		2
3		TBAA	29,700	120	0.024		2
4		TCAA	32,600	600	0.0013		2
5		TCAA	200	10	0.5	2,800 g L ⁻¹ Fe ⁰	3
6		DCAA	40	10	0.4	900 g L ⁻¹ BAC ^a	3
7		MCAA	76	10	0.4	900 g L ⁻¹ BAC ^a	3
8		DCAA	5,805	4500	0.0003	2.4 g L ⁻¹ Fe ⁰	4
9		BAA	9,710	10	0.006		4
10		DBAA	9,803	15	0.14		4
11		TBAA	4,451	10	0.34		4

12	Photochemical system	MCAA	189	3	0.12	1.0 mM S(IV)	5
13		MCAA	4,253	3	0.061	3.73×10^{-6} einstein $L^{-1} s^{-1}$ UV	5
14		MCAA	20,000	50	3.996	3.13×10^4	6
15		DCAA	20,000	30	9.738	$\mu\text{mol m}^{-2} s^{-1}$ UV	6
16		TCAA	20,000	25	13.902		6
17		TCAA	1,000	90	0.042	0.34 g L^{-1} F- TiO_2^b 12 V/V% MeOH ^c $87 \mu\text{W cm}^{-2}$ UV	7
18	Electrocatalysis	DBAA	1,000	0.183	9.16	-1.5 V/SHE using REM ^d	8
19		DBAA	1,000	0.183	33.3	-1.5 V/SHE ^e on WCNT- REM ^d	8
20		DBAA	1,000	0.183	18.3	-1.5 V/SHE using PAC- REM ^d	8
21		TCAA	500	40	0.041	-0.5 V using Pd/rGO/CFP ^f	9
22		TCAA	500	20	0.137	-1.2 V/SCE ^g on the GR-Cu foam ^h	10

23		TCAA	500	30	0.0613	0.9 mA cm ⁻²	¹¹
24		TCAA	500	10	0.5802	over Pd-In/Al ₂ O ₃	¹¹
25		TCAA	500	100	0.0283	-1.2 V/SCE over ANP electrode ⁱ	¹²
26		TCAA	500	100	0.0184	-1.2 V/SCE on Pd/C electrode	¹²
27		TCAA	5,000	60	1.48	-0.5 V/RHE on	¹³
28		TCAA	5,000	60	0.091	CCC/Pd ^j with visible light	¹³
29		TCAA	500	10	0.271	-1.2 V at NG-Cu foam cathode ^k	¹⁴
30		TCAA	817,000	150	0.0087	70 mA on the Pd-loaded Ni cathode	¹⁵
31		DCAA	180	6.66667E-05	51000	1.0 mM S(IV) Fe/GO membrane	This work

^a BAC indicates biologically active carbon processes.

^b F-TiO₂ represents doping fluoride (F) on TiO₂.

^c V/V indicates the volume ratio of methanol over water.

^d REM, MWCNT-REM, and PAC-REM represent Ti_4O_7 reactive electrochemical membranes, multiwalled carbon nanotubes- Ti_4O_7 composite reactive electrochemical membranes, and powder activated carbon- Ti_4O_7 composite reactive electrochemical membranes, respectively.

^e SHE indicates a standard hydrogen electrode.

^f Pd/rGO/CFP indicates a Pd/reduced graphene oxide hybrid catalyst fabricated on carbon fiber paper.

^g SCE represents a saturated calomel electrode.

^h GR-Cu foam represents a three-dimensional graphene-copper (3D GR-Cu) foam electrode.

ⁱ ANP indicates amorphous nickel phosphide.

^j CCC/Pd represents a Cu/Cu₂O/CuO electrode.

^k NG-Cu foam represents a noble metal-free N-doped graphene-Cu (NG-Cu) foam electrode.

7. Supplementary Table 2 compares the permeance and removal efficiency of the Fe/GO membrane with other nanofiltration and reverse osmosis membranes. How is the process of the Fe/GO membrane defined? Is it a nanofiltration membrane?

Response/Action:

The Fe/GO membrane and the related process for DCAA removal are defined as catalytic membrane and membrane catalysis, respectively (Supplementary Figure 18). When the feed water passes through the small channels of the membrane, haloacetic acids (HAAs) are not strongly adsorbed or retained, but instead reduced to nontoxic products by reactive reducing radicals. Therefore, under ideal conditions (without the membrane loss), the Fe/GO membrane system can

continuously produce clean permeate water that undergoes only one pass through the catalytic membrane. Although the removal mechanisms of Fe/GO membrane are different from those of nanofiltration, reverse osmosis, and forward osmosis membranes, a comparison with those membranes can provide insights into the performance (i.e., water permeance and removal efficiency) of Fe/GO membrane and its potential for practical applications.

8. The clarity of the figures is crucial for understanding the results. In Fig. 3a, where two sets of data are presented (water permeance and DCAA reduction efficiency), it is important to clarify the correspondence between the columns/scatter plots and the two parameters to aid interpretation. Similar issue exists in several other figures.

Response/Action:

Following the reviewer's suggestion, we have carefully revised the figures, including the columns/scatter plots, to further clarify the data presented. The following revision has been incorporated:

Fig. 3 Performance of Fe/GO membranes. a, Water permeance and DCAA reduction efficiency of Fe/GO membranes as a function of iron loading amounts. Conditions of the feed solution: initial DCAA level = 180 $\mu\text{g L}^{-1}$, initial S(IV) level = 1.0 mM, pH_{ini} 7.0 \pm 0.1, and 25 \pm 0.5 $^{\circ}\text{C}$ g, Stability test of water permeance and reduction efficiency over cycle times.

9. Regarding the experiments for Figure 3d, it is essential to provide details on how they were conducted. Specifically, what methodology was employed to determine the concentration of DCAA within 1 ms? This information is critical for understanding the experimental setup and the rapid kinetics involved in the dechlorination process.

Response/Action:

We thank the reviewer for his/her comments. The details on how to determine the retention time have been included in the new caption of Figure 3d in the revised manuscript. For instance, the retention time of 1 ms could be achieved by adjusting the applied pressure. The more detailed information was added in the methodology section.

“Fig. 3d, Normalized concentration of DCAA (C/C₀) versus membrane retention time. Retention time indicates the ratio of the pore volume within the catalytic membrane over the flow rate through the membrane. As shown in equation 1 in the methodology according to a previous study⁴⁰, it can be experimentally controlled by adjusting the applied pressure.”

Lines 422–426 *“The retention time can be calculated by equation 1:*

$$t = \frac{V}{Q} = \frac{S \times h}{J \times S \times P} = \frac{h}{J \times P} \quad (1)$$

Where V is the pore volume of membrane (cm³), Q is the flow rate (mL min⁻¹), S (m²) is the effective membrane filtration area, h (nm) is the effective thickness of interlayer free space, and J (L m⁻² h⁻¹ bar⁻¹) and P (bar) are water permeance and applied pressure, respectively.”

Reviewer #2 (Remarks to the Author):

Comments:

The present manuscript reported Ultrafast complete dechlorination enabled by ferrous oxide/graphene oxide catalytic membranes via nanoconfinement advanced reduction. The authors constructed Catalytic membranes by graphene oxide nanosheets with integrated ultrafine and monodisperse sub-5 nm nanoparticles, and found The catalytic membrane achieves ultrafast and complete dechlorination of $180 \mu\text{g L}^{-1}$ dichloroacetic acid to chloride enabled by sulfite radicals, with nearly 100% reduction efficiency within a record-breaking 3.9 ms, accompanied by a six to seven orders of magnitude greater first-order rate constant of $51,000 \text{ min}^{-1}$ than current catalytic processes. This sounds to be interesting. However, I could not recommend it to be accepted for publication by this journal because its novelty is not well documented. Some of my concerns are given below.

Response/Action:

We sincerely appreciate the constructive comments provided by the reviewer. Furthermore, we have carefully prepared detailed point-to-point responses to address each of the reviewer's comments below.

1. The novelty is not well documented. The apparent catalytic performance of the FeO/GO membrane is much higher than the FeO/GO dispersions. The reasons for this great difference are not well clarified. It is necessary to use the nanoconfinement effect to quantitatively explain the

huge difference. In both the abstract, and the introduction, it is not possible to understand the driving force for the catalytic dechlorination.

Response/Action:

We thank the reviewer for his/her constructive comments. We think there are three main reasons for the great difference between Fe/GO dispersions and Fe/GO membranes.

1) Enhanced reactants accessibility: Nanoconfinement effects enhance reactants accessibility to catalysts. For the conventional catalysis in bulk solution, only the contaminants extremely close to catalysts can be reduced (Fig. R1). In contrast, during catalysis based on the novel Fe/GO membranes, all the contaminants have to pass through the nanoconfined transport channels between adjacent GO nanosheets loaded with high-density Fe NPs (Fig. R1). Therefore, the accessibility of reactants to catalytic sites will be dramatically increased, which may be beneficial to decomposition.

Fig. R1. Schematics of processes of catalysis in solution and in membrane.

2) Increased catalyst/reactant ratios: Nanoconfinement effects increase catalyst-to-reactant ratios. According to the detailed calculation in Supplementary Text 1, the molar ratios of Fe to sulfite and Fe to DCAA in the Fe/GO membrane catalysis were in the range of 1168–292 and

418960–104740, respectively. These values are much greater than catalysis in bulk solution (i.e., 0.1–0.5 and 5–20, respectively (*Chem Eng J* **417**, 129115 (2021); *J Hazard Mater* **418**, 125940 (2021); *J Hazard Mater* **384**, 121497 (2020); *Chemosphere* **196**, 593-597 (2018)). Thereby, the ultrahigh ratios of Fe to sulfite, Fe to DCAA, and sulfite to DCAA at confined interlayer spaces of membranes are beneficial to enhanced catalytic performance.

“Supplementary Text 1. Calculation of molar ratios of Fe to sulfite and Fe to DCAA.

In this study, the Fe_{c1.5}/GO membrane had C, O, and Fe atomic contents of 75.98%, 23.27, and 0.75%, respectively. The Fe mass ratio of Fe_{c1.5}/GO was 0.0326 g g⁻¹. For membrane catalytic systems, including DCAA/sulfite solution and Fe_{c1.5}/GO membrane, the molar ratios of Fe to sulfite and Fe to DCAA were calculated as 0.000234 and 0.168, respectively. However, for membrane catalysis, the catalyst and reactants are confined in the transport channels of membranes. Therefore, the molar ratios in reaction regions in membranes are totally different from those in the bulk systems. As demonstrated in previous study¹, the GO membranes could adsorb water and salt solutions with ratios of 1–4 g g⁻¹. Considering the sulfite concentration of 1 mM and the sulfite rejection of 50% through the Fe_{c1.5}/GO membrane, as well as the DCAA concentration of 180 μg L⁻¹ and the limited rejection of membrane for DCAA (<2%), we could calculate that the sulfite and DCAA mass ratios to GO-adsorbed water were 4.1×10⁻⁵ and 1.8×10⁻⁷ g g⁻¹, respectively. Combining the two contents of GO-adsorbed solution and GO-loaded Fe, the molar ratios of Fe to sulfite and Fe to DCAA in the Fe_{c1.5}/GO membrane were estimated to be 1168–292 and 418960–104740, respectively. The ultrahigh ratios of Fe to sulfite, Fe to DCAA, and sulfite to DCAA at confined interlayer spaces of membranes are beneficial to enhanced catalytic performance.”

3) Reduced energy barrier: Nanoconfinement effects reduce energy barrier. Nanoconfinement effects for adjacent GO nanosheets will significantly reduce the energy barrier and facilitate the removal of contaminants as discussed in the section of “*Mechanisms of confinement catalysis and high permeability*”. As shown in Fig. 4a, the activation process of S(IV) involved three stages, with the generation of sulfite radicals ($\text{SO}_3^{\bullet-}$) through the decomposition of FeSO_3^+ being the rate-limiting step. The kinetic free-energy barrier for the dissociation of FeSO_3^+ into $\text{SO}_3^{\bullet-}$ was significantly lower (from 0.41 to 0.86 eV) for Fe/GO membranes compared to FeO NPs (from -1.44 to 0.86 eV), resulting in higher formation of $\text{SO}_3^{\bullet-}$ for the former. Furthermore, Fe/GO required a lower kinetic free-energy barrier for the rate-limiting step of the conversion of DCAA to MCAA, from -0.57 eV to 1.67 eV, compared to FeO NPs (from -0.49 eV to 2.02 eV) in Fig. 4c. Therefore, confinement effects remarkably reduced the energy barriers for the rate-limiting steps, i.e., FeSO_3^+ dissociation into $\text{SO}_3^{\bullet-}$ and DCAA transformation into MCAA, favoring the reduction of DCAA and reaction intermediates through Fe/GO.

Fig. 4 Calculated reaction mechanism of confinement catalysis and superior permeability. *a*, Energy profiles of S(IV) dissociation through Fe/GO and FeO. Chemical structures of FeO, FeH₂SO₃⁺, and FeSO₃⁺ are presented in the first column, while Fe/GO, FeH₂SO₃⁺, and FeSO₃⁺ are displayed in the second column. *c*, Energy profiles of DCAA degradation in Fe/GO, FeO,

GO, and a free environment in the presence of S(IV). In the first column, the structures are presented for FeO, DCAA and MCAA bound to FeO. In the second column, the structures are displayed for GO, DCAA and MCAA bound to rGO. In the third column, the structures of Fe/GO, DCAA and MCAA bound to Fe/GO are presented.

Additionally, we have revised the manuscript. The discussion about the driving force for the catalytic dechlorination has been added in the Abstract, the Introduction, and the Results.

Lines 20–22 “*Benefiting from nanoconfinement effects for reducing energy barriers, enhancing reactants accessibility to catalysts, and increasing catalyst-to-reactant ratios, the catalytic membrane achieves ultrafast and complete dechlorination of 180 $\mu\text{g L}^{-1}$ dichloroacetic acid to chloride enabled by sulfite radicals*”

Lines 61–69 “*We, thus, posit designing ultrafine iron-based NP/graphene oxide composite membranes for advanced reduction of HAAs, which provide nanoconfinement effects to achieve reduced activation energy for the formation of reducing radicals for DCAA dechlorination. Furthermore, the reductants, including S(IV) and contaminants, must undergo nanoconfined transport through the catalyst-loaded interlayer channels of the iron-based NP/graphene oxide composite membranes. This greatly enhances the reductant-catalyst contact probability compared to the bulk solution based catalysts. Additionally, the catalysts and reactions will be nanoconfined in the interlayer transport nanochannels of GO membranes, which will substantially increase the catalyst-to-reactant ratios in confined reaction regions and then promote contaminant degradation.*”

Lines 75–77 “..... which can be attributed to enhanced reactants accessibility to catalytic sites, increased catalyst-to-reactant ratios in reaction regions, and reduced activation energy induced by nanoconfinement effects for FeSO_3^+ dissociation into sulfite radicals and for DCAA dechlorination.”

Lines 249–262 “**Mechanisms of confinement catalysis and high permeability.** Three main reasons can be utilized to explain ultrahigh reduction efficiency. Firstly, nanoconfinement effects enhance reactants accessibility to catalysts. During the Fe/GO membrane catalysis, to reach the permeate side, all reactants including S(IV) and DCAA had to pass the nanoconfined transport channels between adjacent GO nanosheets, which had been loaded with high-density Fe NPs, consequently largely increasing reactants accessibility to catalysts for accelerating DCAA decomposition. Secondly, nanoconfinement effects increase catalyst-to-reactant ratios. All Fe NPs were loaded in the GO interlayer nanochannels, the reaction regions were nanoconfined as well, thereby substantially increasing catalyst-to-reactant ratios in reaction regions for promoting the contaminant degradation. In fact, molar ratios of Fe to sulfite and Fe to DCAA reached as high as 1168–292 and 418960–104740, respectively (see detailed calculation in Supplementary Text 1), which were much higher than those for catalysis in bulk solution (i.e., 0.1–0.5 and 5–20, respectively^{15, 16, 46, 47}). Thirdly, nanoconfinement effects greatly reduce energy barrier. Nanoconfinement effects for adjacent GO nanosheets would significantly reduce the energy barrier and facilitate the removal of contaminants, which was demonstrated by density function theory (DFT) simulation as below.”

Lines 317–319 “*Because of enhanced reactants accessibility to catalytic sites, increased catalyst-to-reactant ratios in reaction regions, and promoted activation of FeSO_3^+ to sulfite radical and conversion of DCAA to MCAA, the membranes show great reduction efficiency for degradation of DCAA to chloride*”

2. **There are some mistakes in Figure 1i**, and the Fe XPS spectra should be re-analyzed. Why FeO NPs were obtained by adding FeCl_3 solution to the GO suspension? The distribution weights of Fe(II) and Fe(III) should be carefully explained. It is also necessary to explain the contributions of Fe(II) and Fe(III) components to the catalytic activity.

Response/Action:

To address the reviewer's concern, we have revised Figure 1i and Supplementary Fig. 13. The revisions are incorporated into the manuscript.

Lines 109–112 “*In addition to the two main peaks of C and O in the XPS spectrum, a new Fe peak was observed in the XPS spectrum of $\text{Fe}_{0.5}/\text{GO}$, with the atomic content of 0.75%. High-resolution XPS spectra indicated that Fe existed as FeO and Fe_2O_3 , with contents of 53.6% and 46.4%, respectively (Fig. 1i).*”

Fig. 1i, Fe 2p XPS spectra of GO and Fe_{c1.5}/GO. The proportions of Fe(II) and Fe(III) were estimated based on the peak areas.

Supplementary Fig. 13. XPS analysis of Fe/GO membranes before and after the stability test. The proportions of Fe(II) and Fe(III) were estimated by calculating the ratio of their respective peak areas to the total peak area of iron. These values have been added to the figures for better clarity.

For synthesis of Fe NPs on GO nanosheets, “We synthesized the Fe/GO composite nanosheets through in-situ growth of FeO NPs by introducing a 1.5 mmol L⁻¹ FeCl₃ solution to the GO

suspension and reduction of Fe^{3+} by NaBH_4 . The electronegative oxygen functional groups of GO adsorbed Fe^{3+} ions, and then FeO NPs would be preferentially and heterogeneously nucleated at active sites under NaBH_4 reduction. FeO NPs were obtained by introducing NaBH_4 to reduce the Fe^{3+} ions that are adsorbed onto the GO.” For more detailed information, please refer to the revised manuscript (Lines 92–96).

We have explained the distribution weights of Fe(II) and Fe(III). The contents of FeO and Fe_2O_3 in Fe NPs of $\text{Fe}_{c1.5}/\text{GO}$ were 53.6% and 46.4%, respectively (Fig. 1i). Furthermore, we have analyzed the contributions of Fe(II) and Fe(III) components to the catalytic activity and revised the relevant parts of the manuscript.

Lines 111 and 112 “High-resolution XPS spectra indicated that Fe existed as FeO and Fe_2O_3 , with contents of 53.6% and 46.4%, respectively (Fig. 1i).”

Lines 234–241 “Less than 2.3% Fe(II) in the $\text{Fe}_{c1.5}/\text{GO}$ membrane was converted into Fe(III) after operation for 20 hrs (Supplementary Fig. 13), indicating the excellent stability of catalytic sites and the efficient conversion of Fe(II)/Fe(III). Reportedly, Fe(II) exhibits a greater ability for activating S(IV) compared to Fe(III)^{21–25}; specially, Fe(II) provides electrons to facilitate the formation of Fe(III)-sulfite complexes, and decomposition of these complexes results in the production of Fe(II), with Fe(III) being the immediate that complexes with sulfite. The relatively stable distribution weight of Fe(II) through facilitated electron shuttling likely by the electrically conductive rGO, further promoted S(IV) activation and DCAA reduction.”

3. Pages 3-4: “membranes with higher FeO loadings appeared darker due to the GO reduction”:

Why was GO reduced?

Response/Action:

During fabrication of Fe/GO composite nanosheets, NaBH₄ was added to reduce the Fe³⁺ to Fe²⁺. Therefore, the GO was reduced as well. To further address the reviewer’s comment, we have removed “due to the GO reduction” to avoid confusion.

Lines 143 “*membranes with higher FeO loadings appeared darker*”

4. Pages 3-4, “which might be beneficial to immobile the mass transfer channels during filtration under pressure and then improve permeation properties.”: These channels are substantially not in the direction of the pressure flow of water.

Response/Action:

We thank the reviewer for his/her comments. In order to pass through whole GO membranes, the molecules have to pass through horizontal interlayer nanochannels between adjacent nanosheets and vertical defects/edges (Fig. R1). Interlayer nanochannels are dominant transport pathways, due to the high width-to-thickness ratio of nanosheets. There are some wrinkles appeared on the GO membranes, which implied that the nanosheets are flexible. During pressure-driven filtration, the flexible GO membranes with wrinkles would be compressed, thereby inducing the reduction in permeation property (*Carbon* **108**, 568-575 (2016); *ACS Nano* **12**, 9309-9317 (2018); *Nat*

Commun **15**, 164 (2024)). For Fe/GO membranes, the wrinkles were weakened due to the reduced flexibility from the FeO growth of nanosheets. This increased stiffness and decreased wrinkles might be beneficial in keeping permeation properties under pressure. We have revised the sentence of “*which might be beneficial to.....improve permeation properties*” to “*which might be beneficial to.....maintain permeation properties*” (Lines 147 and 148).

Fig. R1. Schematics of processes of catalysis in membrane.

5. The use of S(IV) with initial level of 1.0 mM should be given as early as possible.

Response/Action:

The initial concentration of S(IV) (1.0 mM) has been included and can be found in the revised manuscript.

Lines 189 and 190 “*We evaluated the performance of the membranes for DCAA degradation in the presence of 1.0 mM S(IV).*”

6. The system of Fe-oxide NPs + GO + S(IV) should be added as a control in Fig. 3.

Response/Action:

To address the reviewer's concern, we conducted additional control experiments on the removal of DCAA by the system of FeO NPs + GO + S(IV). The FeO NPs was filtrated on GO membranes and then used for DCAA removal. The results are presented in Fig. 3. We have added the relevant discussion in the revised manuscript (Lines 207–213).

“Furthermore, an additional FeO/GO membrane was synthesized by filtering commercial FeO nanoparticles (20 nm) onto the GO membrane. This composite membrane showed a removal efficiency of 45.6% in the presence of 1.0 mM S(IV) (Fig. 3c). However, the ultrafast DCAA degradation was only observed in Fe/GO membrane catalytic systems, where Fe NPs were loaded in the interlayer spaces of GO, indicating that FeO needs to be placed within a confined space to enhance contact between reactants with FeO, and to decrease energy barriers associated with S(IV) activation and DCAA reduction for achieving unprecedented degradation performance.”

Fig. 3c, Reduction efficiency of DCAA under various processes. Conditions of the feed solution: initial DCAA level = 180 $\mu\text{g L}^{-1}$, $\text{pH}_{\text{ini.}}$ 7.0 ± 0.1 , and 25 ± 0.5 °C.

7. Fig.3e summarized the first-order rate constant for reducing DCAA, but the reaction conditions should be listed in detail such as catalysts dosage, S(IV) concentration and so on.

Response/Action:

Following the reviewer’s suggestion, we have included detailed information on reaction conditions in the revised manuscript, such as catalyst doses, S(IV) concentrations, and initial concentrations of HAAs, in the Table 1 of the revised Supporting Information.

Supplementary Table 1. First-order rate constants and corresponding experimental conditions for HAAs removal in this study and previous studies.

No.	Processes	Types of HAAs	Conc. ($\mu\text{g L}^{-1}$)	Reaction time (min)	Rate (min^{-1})	Remarks	Ref.
1	Chemical reduction in bulk solutions	BDCAA	29,120	15	0.177	13.9 $\text{g L}^{-1} \text{Fe}^0$	2
2		CDBAA	20,800	120	0.024		2
3		TBAA	29,700	120	0.024		2
4		TCAA	32,600	600	0.0013		2
5		TCAA	200	10	0.5	2,800 $\text{g L}^{-1} \text{Fe}^0$	3
6		DCAA	40	10	0.4	900 $\text{g L}^{-1} \text{BAC}^a$	3
7		MCAA	76	10	0.4	900 $\text{g L}^{-1} \text{BAC}^a$	3
8		DCAA	5,805	4500	0.0003	2.4 $\text{g L}^{-1} \text{Fe}^0$	4

9		BAA	9,710	10	0.006		4
10		DBAA	9,803	15	0.14		4
11		TBAA	4,451	10	0.34		4
12	Photochemical system	MCAA	189	3	0.12	1.0 mM S(IV)	5
13		MCAA	4,253	3	0.061	3.73×10^{-6} einstein $L^{-1} s^{-1}$ UV	5
14		MCAA	20,000	50	3.996	3.13×10^4	6
15		DCAA	20,000	30	9.738	$\mu\text{mol m}^{-2} s^{-1} \text{ UV}$	6
16		TCAA	20,000	25	13.902		6
17		TCAA	1,000	90	0.042	0.34 g L^{-1} F- TiO_2^b 12 V/V% MeOH ^c $87 \mu\text{W cm}^{-2} \text{ UV}$	7
18		DBAA	1,000	0.183	9.16	-1.5 V/SHE using REM ^d	8
19	DBAA	1,000	0.183	33.3	-1.5 V/SHE ^e on WCNT- REM ^d	8	
20	DBAA	1,000	0.183	18.3	-1.5 V/SHE using PAC- REM ^d	8	
21	TCAA	500	40	0.041	-0.5 V using Pd/rGO/CFP ^f	9	

22		TCAA	500	20	0.137	-1.2 V/SCE ^g on the GR-Cu foam ^h	¹⁰
23		TCAA	500	30	0.0613	0.9 mA cm ⁻² over Pd-	¹¹
24		TCAA	500	10	0.5802	In/Al ₂ O ₃	¹¹
25		TCAA	500	100	0.0283	-1.2 V/SCE over ANP electrode ⁱ	¹²
26		TCAA	500	100	0.0184	-1.2 V/SCE on Pd/C electrode	¹²
27		TCAA	5,000	60	1.48	-0.5 V/RHE on	¹³
28		TCAA	5,000	60	0.091	CCC/Pd ^j with visible light	¹³
29		TCAA	500	10	0.271	-1.2 V at NG- Cu foam cathode ^k	¹⁴
30		TCAA	817, 000	150	0.0087	70 mA on the Pd-loaded Ni cathode	¹⁵
31		DCAA	180	6.66667E- 05	51000	1.0 mM S(IV) Fe/GO membrane	This work

^a BAC indicates biologically active carbon processes.

^b F-TiO₂ represents doping fluoride (F) on TiO₂.

^c *V/V indicates the volume ratio of methanol over water.*

^d *REM, MWCNT-REM, and PAC-REM represent Ti₄O₇ reactive electrochemical membranes, multiwalled carbon nanotubes-Ti₄O₇ composite reactive electrochemical membranes, and powder activated carbon-Ti₄O₇ composite reactive electrochemical membranes, respectively.*

^e *SHE indicates a standard hydrogen electrode.*

^f *Pd/rGO/CFP indicates a Pd/reduced graphene oxide hybrid catalyst fabricated on carbon fiber paper.*

^g *SCE represents a saturated calomel electrode.*

^h *GR-Cu foam represents a three-dimensional graphene-copper (3D GR-Cu) foam electrode.*

ⁱ *ANP indicates amorphous nickel phosphide.*

^j *CCC/Pd represents a Cu/Cu₂O/CuO electrode.*

^k *NG-Cu foam represents a noble metal-free N-doped graphene-Cu (NG-Cu) foam electrode.*

8. Is there other radicals in the system except the SO₃•-, such as active hydrogen atoms (H*) or electrons.

Response:

Based on the EPR spectra results in Figure 4b, it can be concluded that sulfite radicals were the primary reducing active species in Fe/GO membrane catalytic systems; other reactive reducing radicals, such as hydrogen atom radicals and hydrated electrons, were negligible in this membrane catalytic system.

Fig. 4b, EPR spectra of the 5,5-dimethyl-1-pyrrolidine-N-oxide (DMPO) spin-trapping for $Fe_{c.l.s}/GO$ membrane catalytic systems and $FeO/S(IV)$ bulk solutions. Circle symbols indicate signals of the $DMPO/SO_3^{\bullet-}$ adduct. Conditions: DMPO concentration = 100 mM, initial $S(IV)$ level = 1.0 mM, FeO dosage = 1.0 g L^{-1} , pH_{ini} 7.0 \pm 0.1, and 25 \pm 0.5 $^{\circ}C$.

9. The ultra-fast dechlorination of DCAA on Fe/GO membrane was considered to be attributed to the nanoconfinement effects from GO nanosheets from your DFT calculations. Why the dechlorination efficiency is not well by using Fe/GO nanosheets as catalysts in bulk solutions (Supplementary Figure 10)? Why the catalytic activity could increase orders of magnitudes on the Fe/GO membrane?

Response:

We thank the reviewer for his/her constructive comments. As mentioned in comment #1, there are three main reasons for the ultra-fast dechlorination of DCAA on Fe/GO membrane.

1) Enhanced reactants accessibility. For Fe/GO membrane, the interlayer channels are the dominant transport pathways. Molecules, such as S(IV) and DCAA, have to pass through the catalyst-loaded interlayer channels between adjacent GO nanosheets, to reach the permeate sides. It enhanced the accessibility of reactants to catalysts and accelerated the S(IV) activation and DCAA reduction.

2) Increased catalyst/reactant ratios. As mentioned in response of comment #1, the Fe/GO membrane catalysis provides ultrahigh concentration ratios of Fe to sulfite and Fe to DCAA compared to catalysis in bulk solution. The ultrahigh Fe to sulfite, Fe to DCAA, and sulfite to DCAA ratios at confined interlayer spaces of membranes contribute to the ultrahigh performance as well.

3) Reduced energy barrier. As demonstrated in the section of “Mechanisms of confinement catalysis” and mentioned in response of comment #6, nanoconfinement effects will significantly reduce the energy barrier, and facilitate the S(IV) activation and DCAA decomposition.

Lines 20–22 “*Benefiting from nanoconfinement effects for reducing energy barriers, enhancing reactants accessibility to catalysts, and increasing catalyst-to-reactant ratios, the catalytic membrane achieves ultrafast and complete dechlorination of 180 $\mu\text{g L}^{-1}$ dichloroacetic acid to chloride enabled by sulfite radicals*”

Lines 61–69 “*We, thus, posit designing ultrafine iron-based NP/graphene oxide composite membranes for advanced reduction of HAAs, which provide nanoconfinement effects to achieve reduced activation energy for the formation of reducing radicals for DCAA dechlorination. Furthermore, the reductants, including S(IV) and contaminants, must undergo nanoconfined*

transport through the catalyst-loaded interlayer channels of the iron-based NP/graphene oxide composite membranes. This greatly enhances the reductant-catalyst contact probability compared to the bulk solution based catalysts. Additionally, the catalysts and reactions will be nanoconfined in the interlayer transport nanochannels of GO membranes, which will substantially increase the catalyst-to-reactant ratios in confined reaction regions and then promote contaminant degradation.”

Lines 75–77 “..... which can be attributed to *enhanced reactants accessibility to catalytic sites, increased catalyst-to-reactant ratios in reaction regions, and reduced activation energy induced by nanoconfinement effects* for FeSO_3^+ dissociation into sulfite radicals and for DCAA dechlorination.”

Lines 207–213 “*Furthermore, an additional FeO/GO membrane was synthesized by filtering commercial FeO nanoparticles (20 nm) onto the GO membrane. This composite membrane showed a removal efficiency of 45.6% in the presence of 1.0 mM S(IV) (Fig. 3c). However, the ultrafast DCAA degradation was only observed in Fe/GO membrane catalytic systems, where Fe NPs were loaded in the interlayer spaces of GO, indicating that FeO needs to be placed within a confined space to enhance contact between reactants with FeO, and to decrease energy barriers associated with S(IV) activation and DCAA reduction for achieving unprecedented degradation performance.*”

Lines 249–262 “*Mechanisms of confinement catalysis and high permeability. Three main reasons can be utilized to explain ultrahigh reduction efficiency. Firstly, nanoconfinement effects*

enhance reactants accessibility to catalysts. During the Fe/GO membrane catalysis, to reach the permeate side, all reactants including S(IV) and DCAA had to pass the nanoconfined transport channels between adjacent GO nanosheets, which had been loaded with high-density Fe NPs, consequently largely increasing reactants accessibility to catalysts for accelerating DCAA decomposition. Secondly, nanoconfinement effects increase catalyst-to-reactant ratios. All Fe NPs were loaded in the GO interlayer nanochannels, the reaction regions were nanoconfined as well, thereby substantially increasing catalyst-to-reactant ratios in reaction regions for promoting the contaminant degradation. In fact, molar ratios of Fe to sulfite and Fe to DCAA reached as high as 1168–292 and 418960–104740, respectively (see detailed calculation in Supplementary Text 1), which were much higher than those for catalysis in bulk solution (i.e., 0.1–0.5 and 5–20, respectively^{15, 16, 46, 47}). Thirdly, nanoconfinement effects greatly reduce energy barrier. Nanoconfinement effects for adjacent GO nanosheets would significantly reduce the energy barrier and facilitate the removal of contaminants, which was demonstrated by density function theory (DFT) simulation as below.”

Lines 317–319 “Because of enhanced reactants accessibility to catalytic sites, increased catalyst-to-reactant ratios in reaction regions, and promoted activation of FeSO_3^+ to sulfite radical and conversion of DCAA to MCAA, the membranes show great reduction efficiency for degradation of DCAA to chloride”

Response to the comments from the Reviewers

Reviewer #1 (Remarks to the Author):

Most of the review comments have not been adequately addressed. The experimental remain insufficient, and the calculation of the material ratio unclear. Without these critical data, it is challenging to evaluate the performance of the membrane and, consequently, value of this work.

Response/Action:

We sincerely appreciate the constructive comments provided by the reviewer. Furthermore, we have carefully prepared detailed point-to-point responses to address each of the reviewer's comments below.

1. P3, Line 92: “We synthesized the Fe/GO composite nanosheets through in-situ growth of FeO NPs by introducing a 1.5 mmol L⁻¹ FeCl₃ solution to the GO suspension and reduction of Fe³⁺ by NaBH₄.” What are the concentration and volume of the GO suspension? Were all of the dosed Fe³⁺ completely transformed into FeO and Fe₂O₃? Was any zero-valent iron generated during the reduction with NaBH₄?

Response/Action:

The concentration and volume of the GO suspension are 20 mg L⁻¹ and 150 mL, respectively, as stated in Lines 345–347 of the original manuscript: “Colloidal dispersions of individual GO sheets were prepared in 150 mL of water at a level of 20 mg L⁻¹ using an ultrasonic homogenizer (JY98-IIIDN, Scientz) with ultrasound treatment at 40% power for 2 hrs.”

To response to the reviewer's concern, we have included the following revisions:

Lines 92–94 “We synthesized the Fe/GO composite nanosheets through in-situ growth of FeO NPs by introducing a 1.5 mmol L⁻¹ FeCl₃ solution (20 mL) to the GO suspension (20 mg L⁻¹, 150 mL) and reduction of Fe³⁺ by NaBH₄ (1.6 M, 75 μL).”

Lines 347–350 “A solution of ferric iron ions with a concentration of 0.5–1.5 mmol L⁻¹ (20 mL) was prepared and added to the GO dispersion (20 mg L⁻¹, 150 mL). After adsorption for more than 10 hrs, the mixture was reduced by adding NaBH₄ in a controlled amount (i.e., 1.6 M, 75 μL).”

As shown in Fe 2p XPS spectra (Fig. 1i), almost all the dosed Fe³⁺ was transformed into FeO and Fe₂O₃ after reduction by NaBH₄. No obvious peaks were observed for zero-valent iron.

To address the reviewer's comment, the following revision has been included in the manuscript: Lines 111–113 “High-resolution XPS spectra indicated that Fe existed as FeO and Fe₂O₃, with contents of 53.6% and 46.4%, respectively (Fig. 1i). No obvious peaks were observed for zero-valent iron.”

Fig. 1 Characterizations of Fe/GO nanosheets. i, Fe 2p XPS spectra of GO and Fe_{c1.5}/GO. The proportions of Fe(II) and Fe(III) were estimated based on the peak areas.

2. Supplementary Fig. 17: The x-axis contains three mass data points. Are these the masses of the Fe/GO membranes? The calculation missed some important parameters, such as how many volumes of the feed solution was treated in one hour.

Response/Action:

To avoid potential confusion, we have revised Supplementary Fig. 17 as follows (added x-axis label, with additional explanations included in the figure caption):

Supplementary Fig. 17. Water permeance and DCAA reduction efficiency of Fe/GO membranes as a function of the loading amount of GO. For instance, 400 μg GO is obtained by vacuum-filtration of 20 mL of a 20 mg L⁻¹ GO solution that was loaded with Fe NPs onto the

substrate membrane. Experimental conditions: initial DCAA level = 180 $\mu\text{g L}^{-1}$, initial S(IV) level = 1.0 mM, pH_{ini} 7.0 \pm 0.1, applied pressure = 1.0 bar, membrane area = 8 cm^2 , and temperature = 25 \pm 0.5 $^{\circ}\text{C}$. Error bars represent the standard deviation from at least triplicate experiments. Error bars smaller than symbols are invisible. The volumetric loading of feed solution per hour can be obtained by the product of water permeance, applied pressure, and membrane area. In the current study, the volumetric loadings of feed solution within a filtration duration of 1 hr are 3.67 L, 0.0389 L, and 0.0236 L at GO loadings of 80 μg , 400 μg , and 1600 μg , respectively.

3. “the molar ratio of Fe to sulfite in reaction regions of Fe/GO membranes was ultrahigh” “the molar ratios of Fe to sulfite and Fe to DCAA were calculated as 0.000234 and 0.168.” What do you mean by using the word “ultrahigh”? What is the mechanism in the Fe/GO mediated catalysis?

Response/Action:

A. Molar ratios

We would like to clarify that the “nominal” molar ratios of 0.000234 for Fe to sulfite and 0.168 for Fe to DCAA were calculated based on their mass in the entire membrane system. However, in the confined reaction regions, the effective molar ratios can be as high as 1168–292 for Fe to sulfite and 418963–104741 for Fe to DCAA (see details below). To address the reviewer’s concern, we have revised Supplementary Text 1 and added Supplementary Table 3 as follows:

“Supplementary Text 1. Calculation of molar ratios of Fe to sulfite and Fe to DCAA.

In this study, the $\text{Fe}_{\text{c}1.5}/\text{GO}$ membrane had C, O, and Fe atomic contents of 75.98%, 23.27, and 0.75%, respectively. The Fe mass ratio of $\text{Fe}_{\text{c}1.5}/\text{GO}$ was 0.0326 g g^{-1} . Based on the mass values in entire membrane system (including DCAA/sulfite solution and $\text{Fe}_{\text{c}1.5}/\text{GO}$ membrane), the nominal molar ratios were calculated as 0.000234 for Fe to sulfite and 0.168 for Fe to DCAA (Supplementary Table 3). However, for membrane catalysis, the catalyst and reactants are confined in the transport channels of membranes. Therefore, the effective molar ratios in confined reaction regions of membranes are much higher than the nominal molar ratios in entire membrane system and those for catalysis in bulk solution^{1, 2, 3, 4}. As demonstrated in previous study⁵, the GO membranes could adsorb water from aqueous solutions with ratios of 1–4 g g^{-1} . Considering (a) the sulfite concentration of 1 mM and the sulfite rejection of 50% by the $\text{Fe}_{\text{c}1.5}/\text{GO}$ membrane and (b) the DCAA concentration of 180 $\mu\text{g L}^{-1}$ and the limited rejection of membrane for DCAA (<2%), we could calculate that their effective concentrations in the reaction region are estimated to be 0.5 mM and 180 $\mu\text{g L}^{-1}$, respectively (see footnotes c and d of Supplementary Table 3). Based on the contents of sulfite, DCAA, and Fe within the catalytic membrane, the effective molar ratios of Fe to sulfite and Fe to DCAA in the reaction region are estimated to be 1168–292 and 418963–104741, respectively (Supplementary Table 3). The enriched ratios of Fe to sulfite and Fe to DCAA at confined interlayer spaces of membranes are beneficial for enhancing catalytic performance.”

Supplementary Table 3. Weight ratios in entire membrane system and confined reaction region.

Nominal ratios in entire membrane system ^a					
	Content (g)	Mass ratio (g g ⁻¹)		Molar ratio (mol mol ⁻¹)	
Fe	1.30×10^{-5}				
Sulfite	8.00×10^{-2}	Fe to sulfite	1.62×10^{-4}	Fe to sulfite	2.34×10^{-4}
DCAA	1.80×10^{-4}	Fe to DCAA	7.24×10^{-2}	Fe to DCAA	1.68×10^{-1}
Effective ratios in confined reaction region					
	Content (g)	Mass ratio (g g ⁻¹)		Molar ratio (mol mol ⁻¹)	
Fe	1.30×10^{-5}				
GO	4.00×10^{-4}				
Water	$(4.00-16.0) \times 10^{-4}$ ^b				
Sulfite	$(1.60-6.40) \times 10^{-8}$ ^c	Fe to sulfite	815-204	Fe to sulfite	1168-292
DCAA	$(7.20-28.8) \times 10^{-11}$ ^d	Fe to DCAA	181226-45306	Fe to DCAA	418963-104741

a. For a membrane coupon area of 55.4 cm².

b. Based on a water uptake of 1-4 g g⁻¹ GO.⁵

c. Sulfite content in the reaction region is calculated based on the water uptake by GO and its effective concentration in the catalytic membrane. For sulfite with a rejection of 50% by the membrane, its effective concentration in the reaction region is estimated as half of the bulk concentration, i.e., $(100\%-50\%) \times 1 \text{ mM} = 0.5 \text{ mM}$.

d. DCAA content in the reaction region is calculated based on the water uptake by GO and its effective concentration in the catalytic membrane. Since the rejection of DCAA by the membrane is negligible (< 2%), its effective concentration in the reaction region is approximated by the bulk concentration (180 μg L⁻¹).

B. Mechanism

As mentioned above, the confined reaction zone can greatly increase the effective molar ratios of Fe to sulfite and Fe to DCAA (Supplementary Text 1 and Supplementary Table 3).

During the catalysis, Fe(II) can provide electrons to promote the formation of Fe(III)-sulfite complexes [21-25] and the decomposition of these complexes regenerates Fe(II), with Fe(III) being the immediate that complexes with sulfite. Therefore, the high molar ratios of Fe to sulfite and Fe to DCAA are beneficial to S(IV) complexation and electron transfer, resulting in the activation of S(IV) and reduction of DCAA.

To further address the reviewer's comment, the following changes have been included:

Lines 255-259 “Secondly, nanoconfinement effects increase catalyst-to-reactant ratios (Supplementary Text 1 and Supplementary Table 3). All Fe NPs were loaded in the GO interlayer nanochannels, the reaction regions were nanoconfined as well, thereby substantially increasing catalyst-to-reactant ratios in reaction regions for promoting S(IV) complexation and electron transfer²¹⁻²⁵ and thus enhancing S(IV) activation and contaminant degradation.”

References

- [21] Xiao Q, Yu S. The role of dissolved oxygen in the sulfite/divalent transition metal ion system: Degradation performances and mechanisms. *Chemical Engineering Journal* **417**, 129115 (2021).
- [22] Zhou D, Chen L, Li J, Wu F. Transition metal catalyzed sulfite auto-oxidation systems for oxidative decontamination in waters: A state-of-the-art minireview. *Chemical Engineering Journal* **346**, 726-738 (2018).
- [23] Kraft J, Van Eldik R. Kinetics and mechanism of the iron (III)-catalyzed autoxidation of sulfur (IV) oxides in aqueous solution. 1. Formation of transient iron (III)-sulfur (IV) complexes. *Inorganic Chemistry* **28**, 2297-2305 (1989).
- [24] Grimme S, Antony J, Ehrlich S, Krieg H. A consistent and accurate ab initio parametrization of density functional dispersion correction (DFT-D) for the 94 elements H-Pu. *J Chem Phys* **132**, 154104 (2010).
- [25] Zhang Y, Zhou J, Li C, Guo S, Wang G. Reaction Kinetics and Mechanism of Iron(II)-Induced Catalytic Oxidation of Sulfur(IV) during Wet Desulfurization. *Industrial & Engineering Chemistry Research* **51**, 1158-1165 (2012).

4. Without the aforementioned experimental details, the conclusions drawn in this work are not well supported.

Response/Action:

We appreciate the reviewer's valuable comments, and we have provided additional experimental information as detailed in our responses #1-3.

Reviewer #2 (Remarks to the Author):

The comments have been well responded, and the present version of this manuscript is possible accepted for publication.

Response/Action:

We appreciate the reviewer's high evaluation of our study.